# Boron encapsulated in a liposome can be used for combinational neutron capture therapy

Jiyuan Li[1,6], Qi Sun[2,6], Chuanjie Lu[1], Han Xiao[3], Zhibin Guo[1], Dongban Duan[1], Zizhu Zhang[4,5], Tong Liu[5] & Zhibo Liu ✉ [1,2✉]

Boron neutron capture therapy (BNCT) is an attractive approach to treat invasive malignant tumours due to binary heavy-particle irradiation, but its clinical applications have been hindered by boron delivery agents with low in vivo stability, poor biocompatibility, and limited application of combinational modalities. Here, we report boronsome, a carboranyl-phosphatidylcholine based liposome for combinational BNCT and chemotherapy. Theoretical simulations and experimental approaches illustrate high stability of boronsome. Then positron emission tomography (PET) imaging with Cu-64 labelled boronsome reveals high-specific tumour accumulation and long retention with a clear irradiation background. In particular, we show the suppression of tumour growth treated with boronsome with neutron irradiation and therapeutic outcomes are further improved by encapsulation of chemotherapy drugs, especially with PARP1 inhibitors. In sum, boronsome may be an efficient agent for concurrent chemoradiotherapy with theranostic properties against malignancies.

[1] Beijing National Laboratory for Molecular Sciences, Radiochemistry and Radiation Chemistry Key Laboratory of Fundamental Science, College of Chemistry and Molecular Engineering, Peking University, Beijing 100871, China. [2] Peking University-Tsinghua University Center for Life Sciences, Beijing 100871, China. [3] University of Chinese Academy of Sciences Beijing, Beijing 100049, China. [4] Beijing Nuclear Industry Hospital, Beijing 100045, China. [5] Beijing Capture Tech Co., Ltd, Beijing 102413, China. [6] These authors contributed equally: Jiyuan Li, Qi Sun. ✉email: zbliu@pku.edu.cn

Boron neutron capture therapy (BNCT) is a binary, biochemically targeted radiotherapy[1,2], which provides excellent tumour control over inoperable malignant tumours[3,4]. The therapeutic effect of BNCT is based on the capture reactions that occur when tumour-targeted boron delivery agents are irradiated with thermal neutrons. Therefore, the high-selective delivery of sufficient boron into tumour cells is the key to developing BNCT drugs, yet is still an unmet need.

Boronated tyrosine derivative 4-boronophenylalanine (BPA) is the most widely used boron delivery agent in the clinical setting[5]. However, deficient uptake has been a long-standing problem since the first application of BPA in 1987 as well as other small-molecule based boron delivery agents[3,6]. To overcome this problem, boron-enriched nanocarriers have been intensively studied as alternative candidates for boron delivery agents during the last two decades[7]. Among them, liposomes are considered to be one of the most efficient and clinical-relevant delivery vehicles, yet several drawbacks have hampered liposomes to become practical boron carriers in clinics: (1) liposome structures currently reported are mainly based on boron-enriched small molecules encapsulation strategy, of which the loading capacity is limited and may have cargo leakage to off-tumour tissues (Fig. 1a)[8]; (2) application of unusual boranes which are short of in vivo stability could induce un-expected biochemical toxicity and immunogenicity (Fig. 1b)[9–11]; (3) absence of suitable properties to allow their detection by noninvasive imaging techniques to explore in vivo biodistribution of boron agents to ensure the accuracy of neutron irradiation and to improve efficacy[12].

To meet the above challenge, we report carborane-derived liposome mimics, denoted as boronsome, for imaging-guided chemotherapy-assisted BNCT. Carborane has been a star molecule recently in the practice areas of medicine, nanomaterials and catalysis[13]. Molecular characteristics such as thermal and redox stability, high hydrophobicity and low nucleophilicity give carborane unique advantages in drug development[14]. Not to mention that one carborane molecule contains ten boron atoms, which makes carborane a perfect candidate for BNCT application[15]. To develop biocompatible boronated liposomes with high stability and high boron content, carboranyl group is covalently conjugated to the hydrophobic tail of phospholipids to form a series of BoPs (Supplementary Fig. 1). Thiol-halo reaction was adopted instead of click chemistry to maintain the flexibility of phospholipids. As for the hydrophilic head, the structure of phosphocholine was maintained to be more biocompatible and reduce potential toxicity[16]. In addition, different from previous work which encapsulates small-molecule boron species inside liposomes[8,17], we use boron-containing moiety to form liposome membranes, denoted as boronsomes (Fig. 1c), thus freeing the internal cavity to carry other drugs for combination therapy. In comparison with other reported liposome structures, it turns out that boronsome carries more boron which implies a higher boron content in tumour[18–20].

In this work, we report boronsome as a delivery agent with high biocompatibility and high tumour accumulation and can be used for positron-emission tomography (PET) imaging-guided BNCT with concurrent chemoradiotherapy. Boronsome containing carboranyl phospholipid exhibits high stability in 50% bovine serum for over 3 days. The intracellular uptake of boron in cultured cancer cells incubated with boronsome for 12 h is up to $182\,\mu g/10^6$ cells, 61% higher than BPA. In animal studies, PET imaging with [$^{64}$Cu]Cu-NOTA-boronsome shows long tumour retention (8 h to 48 h p.i.) and low peritumoural normal tissues uptake. Peak T/N ratio is found at 12 h post injection (tumour/muscle = $37 \pm 1$; tumour/brain = $62 \pm 2$; tumour/bone = $17 \pm 1$; tumour/blood = $4 \pm 0.3$). Tumour-to blood ratio, the most essential value in BNCT, stands out from various reported structures showing the high boron delivering efficiency of boronsome. Tumourous boron accumulation (93.3 ppm) meets the therapeutic requirement at 12 h after administration as well, which has an advantage over other reported structures[18,19,21–23]. As expected, tumour growth of mice treated with boronsome followed by neutron irradiation is significantly suppressed than in other control groups (Fig. 1d). Remarkably, BNCT effectiveness is further amplified through DNA repair system interference by PARP1 inhibitor encapsulation. Meanwhile, death patterns switch in cancer cells is observed after BNCT based concurrent chemoradiotherapy. In sum, boronsome holds a great potential to be an efficient and versatile delivery system for BNCT against malignancies in clinics.

## Results

The hydrophobic tails of phospholipids in liposomal bilayer are usually linear or two-dimensional[24], while carborane is a three-dimensional icosahedron. To test whether the steric hindrance caused by carborane inhibits close apposition of bilayers, we synthesised four boronated phospholipids (BoPs) with variation in the length of lipid arms (Fig. 2a), derived from carboxyl acids containing 4, 8, 12 and 16 carbon atoms, respectively. Next, we examined the 3D structure of the BoPs, and qualitatively noticed visible discrepancies among the configurations (Fig. 2b). Several key parameters were taken into quantitatively analysis: $l_c$,

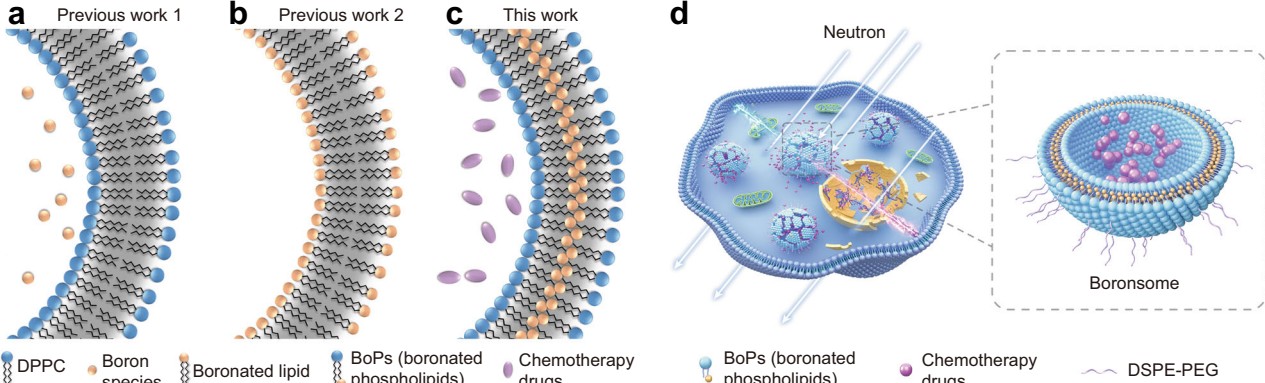

**Fig. 1 Schematic illustration of boronated liposomes for BNCT. a–c** Formulations of previous boronated liposomes (**a**, **b**) and boronated liposomes in this work (boronsomes, (**c**). **d** Boronsomes emit high-energy particles when irradiated by neutrons and release chemotherapy drugs sequentially to kill cancer cells.

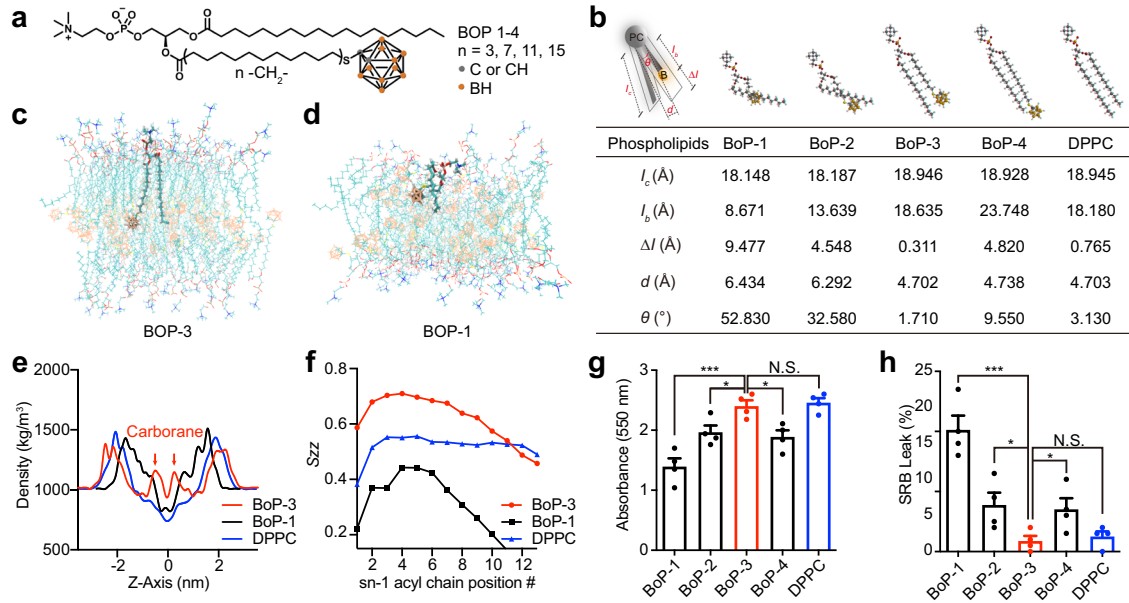

**Fig. 2 Boronated phospholipids (BoPs) with hydrophobic tails of matched length show superior self-assembly and packing properties. a** Chemical structure of BoPs with different lengths of alkyl chain. **b** Spatial configuration and structural parameters of corresponding BoPs. $l_c$, carbon arm length (Å); $l_b$, boron arm length (Å); $\Delta l$, length difference between lipid arms (Å); $d$, distance between lipid arms (Å); $\theta$, dihedral angle between lipid arms (°). **c** Cross-sectional area of bilayers formed from BoP-3 and Dipalmitoylphosphatidylcholine (DPPC) in a 10 ns, 128 monomer MD simulation. Lipids are depicted in wireframe format. **d** Cross-sectional area of bilayers formed from BoP-1 and DPPC in a 10 ns, 128 monomer MD simulation. Lipids are depicted in wireframe format. **e** BoPs and DPPC density (excluding water contribution) post 10 ns MD simulation. **f** Chain order parameter ($S_{ZZ}$) for BoPs following 10 ns MD simulation. $S_{ZZ}$ indicates order of lipid chain with respect to bilayer normal vector. Error bars show range for last two adjacent 10 ns block means. **g** Encapsulation of SRB ($n = 4$) by various BoPs and DPPC under same conditions. ***$p = 0.0010$, BoP-2 vs BoP-3: *$p = 0.0264$, BoP-4 vs BoP-3: *$p = 0.0126$. **h** Release of SRB from boronsomes ($n = 4$) with various BoPs and DPPC in 50% bovine serum at 37 °C after 24 h. ***$p = 0.0003$, BoP-2 vs BoP-3: *$p = 0.0358$, BoP-4 vs BoP-3: *$p = 0.0433$. Three independent experiments were performed and representative results are shown. N.S., no significant difference, data are shown as mean ± SEM, two-tailed unpaired Student's $t$ test. Source data are provided as a Source Data file.

represents the length of palmitoyl lipid arm; $l_b$, represents the length of the carboranyl-lipid arm; $\Delta l$: represents the length difference between lipid arms; $d$: represents the distance between lipid arms; $\theta$: represent the dihedral angle between lipid arms. Compared with DPPC (Dipalmitoylphosphatidylcholine), $l_c$ of each BoPs does not differ much while $l_b$ decreases gradually according to the number of carbon atoms. Additionally, $\Delta l$ of BoP-3 is the smallest, even smaller than that of natural lipid DPPC. It has been reported that it is favourable to form stable nanostructures when the two chains of phospholipids are similar in length[25]. For $d$ and $\theta$, the values of BoP-3 are also the closest to the values of DPPC among all BoPs. As a matter of fact, the reason why there were large gaps between other BoPs and DPPC, especially BoP-1 and BoP-2, is that molecules sacrifice distance and angle between lipid arms in order to make them as similar in length as possible to achieve the lowest-energy configuration. Based on initial analysis, BoP-3 may be the phospholipid of choice to form lipid bilayers with preferable self-assembly and packing features.

MD simulations were proved to be very powerful for molecular properties determination of lipid bilayers[26]. This approach was used to assess the hypothesis that BoP-3 could form nanostructures with better stability and cargo loading efficiency. We performed MD simulations of a bilayer system composed of 128 molecules of BoPs and DPPC at a ratio of 5.4:1 after modification of the carborane-lipid force field (Supplementary Tables 1–3). 3600 water molecules were added to produce a 5 nm layer. Figure 2c shows a cross-section of the BoP-3 bilayer, one of the BoP-3 molecules was shown in bold. Gave out a standard membrane structure with a regular sequence and a low chain-folding deformation degree compared with other BoPs (Fig. 2d,

Supplementary Fig. 2). Through a 10 ns simulation, the bilayer density plot (Fig. 2e) indicated that BoP-3 generates a thicker bilayer (5.24 versus 3.93 nm with the Gibbs–Luzzati criterion) compared to BoP-1 (Supplementary Table 4). The chain order parameter ($S_{ZZ}$) indicates the orientation of the lipid chain with respect to the bilayer normal. Values near 1 indicate an average orientation parallel to the bilayer normal and values closer to 0 indicate an orientation away from bilayer normal. As shown in Fig. 2f, positions of the sn-1-palmitoyl chains in BoP-3 bilayers had less disorder than other lipid bilayers based on an average from 5 to 10 ns. In conclusion, MD simulations demonstrated that BoP-3 bilayers would have enhanced stability and cargo loading property compared with other lipid bilayers as a result of elevated bilayer thickness, enhanced intramolecular hydrogen bonding and improved regularity in the palmitoyl side chains.

The above BoPs have been synthesised to experimentally compare the behaviour of the nanostructures formed by different BoPs or DPPC. Cargo loading efficiency was assessed by Sulphorodamine B (SRB) encapsulation (Fig. 2g). Lipids were hydrated with a 50 mM solution, followed by gel filtration to obtain liposomal formations. Absorbances at 550 nm were found highest in BoP-3 boronsome and close to that of DPPC liposome. SRB release was assessed in 50% bovine serum at 37 °C after 24 hours-incubation to evaluate the stability (Fig. 2h), and BoP-3 boronsome possessed significantly higher retention efficacy than that of other BoPs.

Thereby we proposed BoP-3 boronsome as a metabolically biocompatible stable platform for liposomal boron delivery (Fig. 3a). Followed by size controlling with extruder and purification by dialysis, boronsomes were spherical vehicles of 100 nm in diameter according to the transmission electron

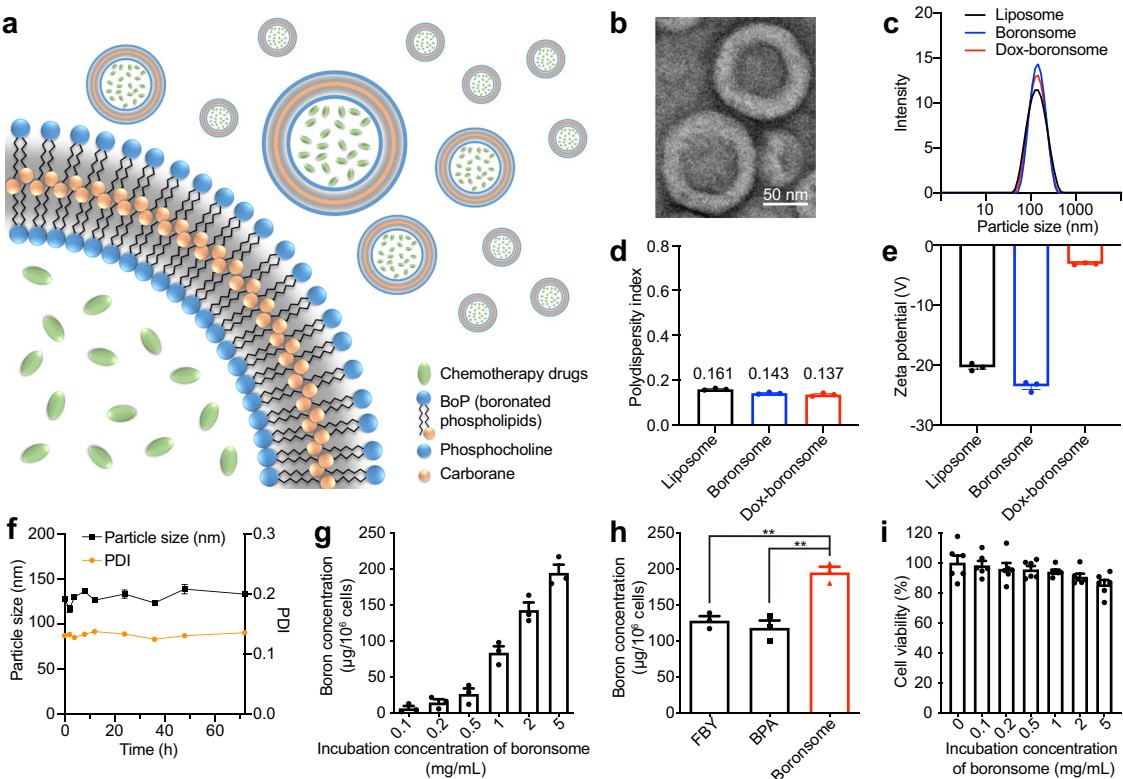

**Fig. 3 Boronsome is an efficient platform for boron and chemotherapy drug delivery. a** Schematic illustration of chemotherapeutic agents encapsulated carboranyl-lipid boronsomes. **b** Representative TEM images of boronsomes. Three independent experiments were performed and representative results are shown. Scale bar, 50 nm. **c–e** Dynamic light scattering size **c**, PDI **d** and zeta potential **e** of liposomes, boronsomes and Dox-boronsomes ($n = 3$). **f** Boronsomal stability in 50% FBS. The size and PDI ($n = 3$) were measured at different time points. **g** Boron concentration ($n = 4$) of 4T1 cells after incubation with various concentrations of boronsome for 24 h. **h** Boron concentration ($n = 4$) of 4T1 cells after incubation with FBY, BPA and boronsomes for 24 h at the same boron dose (two-tailed unpaired Student's $t$ test, FBY vs Boronsome: **$p = 0.0027$, BPA vs Boronsome: **$p = 0.0040$). **i** Survival fraction ($n = 6$) of 4T1 cells under various concentrations of boronsome after 24 h. Data are means ± SEM. Source data are provided as a Source Data file.

microscopy (TEM) measurement (Fig. 3b, Supplementary Fig. 3a). At higher magnifications, the boronsome structure was identified as a bilayer material with a thickness of 15.78 nm. In compliance with unloaded liposomes, Doxorubicin (Dox) loaded boronsomes exhibited uniform morphology and good mono-dispersity as well. Size of Dox-boronsome was similar to both boronsome and liposome, with the individual size around 125 nm and polydispersity index (PDI) of ~0.15, which was in favour of the EPR effect (Fig. 3c, d). Zeta potential of liposomes, boronsomes and Dox-boronsomes were measured (Fig. 3e), which expresses the potential difference between the dispersion medium and the stationary layer of fluid attached to double-layer properties to characterise a realistic magnitude of surface charge[27]. Electric neutralised to −3.1 mV after Dox encapsulation showed efficient cargo loading property. Finally, barely changed curves in both particle size and PDI demonstrated the long-lasting stability of boronsomes in 50% bovine serum at 37 °C (Fig. 3f).

In vitro performance of being a qualified boron delivery agent was tested after boronsome was characterised as a biomimetic stable bilayer vesicle. Cellular boron concentrations were first evaluated by inductively coupled plasma optical emission spectroscopy (ICP-OES). Mouse triple-negative breast cancer 4T1 cells were incubated with boronsomes at concentrations varied from 0.1 to 5 mg/mL for 24 h. Cellular boron concentration gradually increased with dose escalation of boronsome (Fig. 3g). Boron concentration in 4T1 cells was up to 182.5 ppm at the dose of 5 mg/mL which fully meets the demand of clinical BNCT[7]. Higher boron concentration could still be acquired when incubated with a higher dose (Supplementary Fig. 3b). Moreover,

an elevated boron delivery ability of boronsome was shown in a comparison assay with the clinical boron delivery agent BPA and previously reported boronated amino-acid derivatives FBY (Fig. 3h)[28]. These results indicated that nano boron carriers have clear advantages over small molecules in boron delivery. Cytotoxicity was then assessed with a cell counting kit-8 (CCK-8) assay. As shown in Fig. 3i, boronsomes alone exhibited good tolerance when incubation concentration reached 5 mg/mL. However, a slight decline in cell survival fraction was observed when incubated with boronsome at 10 mg/mL (Supplementary Fig. 3c). In sum, boronsome was proved to be a bio-safe and efficient delivery system for boron and chemotherapy drugs.

Noninvasive techniques that can trace boron in real-time are critical for in-patient screening, treatment planning and efficacy evaluation[29]. PET imaging has already been demonstrated powerful in imaging-guided BNCT both in preclinical and clinical studies. Meanwhile, boronsome accumulates in the tumour due to the permeability and retention (EPR) effect, which may be significantly heterogeneous in patients. Therefore, boronsomes labelled with radioactive nuclides would allow the identification before choosing whether or not to be treated with boronsome-BNCT, which could enable precision medicine. We incorporated radiolabelled lipid on the bilayer, enabling PET imaging to quantitatively measure local boron concentration, and thereby precisely determine the desired intensity and the irradiation time for neutron beam[30]. DSPE-PEG2000-NOTA was applied as chelators of positron-emission nuclides Cu-64 for PET imaging[31]. [64Cu]Cu-NOTA-boronsome were purified by gel filtration. Up to 24 h of stability was validated through radio-TLC

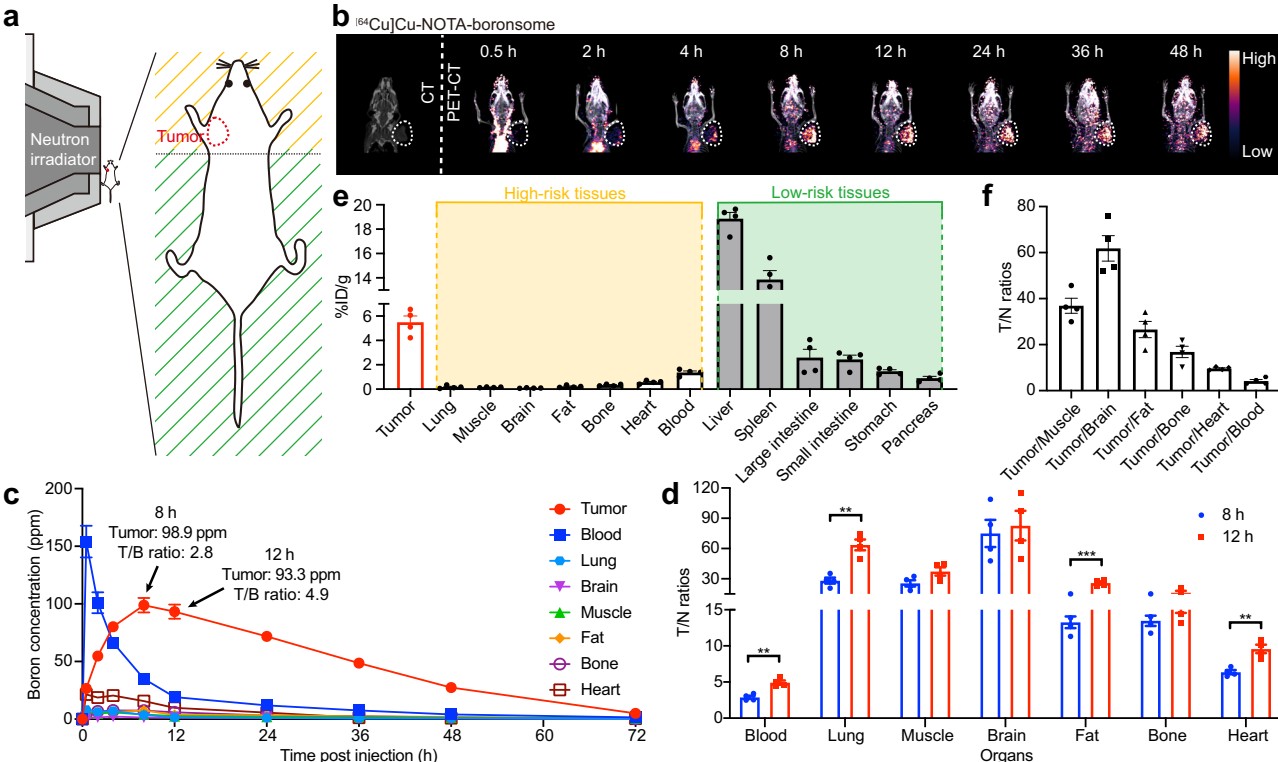

**Fig. 4 PET imaging indicated that boronsome showed high accumulation in tumour but low uptake in other high-risk tissues. a** Schematic representation of a mouse located at the beam terminal of the neutron irradiator. Tissues within radiation radius are marked as high-risk tissues (yellow). Tissues outside the radiation radius are marked as low-risk tissues (green). **b** Representative CT and PET-CT images of the female BALB/c mouse at various time points depicting [$^{64}$Cu]Cu-NOTA-boronsome uptake. Tumours were circled with white dashed lines. **c** Time-dependent boron concentration in tumour, blood, lung, brain, muscle, fat, bone and heart post injection of boronsome (500 mg/kg, i.v.) measured by ICP-OES ($n = 4$). **d** Tumour-to-normal tissue ratios (T/N ratios) of organs 8 and 12 hours post once injection of boronsome (500 mg/kg, i.v., $n = 4$). Two-tailed unpaired Student's $t$ test, blood: **$p = 0.0015$; lung: **$p = 0.0012$; fat: ***$p = 0.0003$; heart: **$p = 0.0026$. **e** Corresponding biodistribution ($n = 4$) of [$^{64}$Cu]Cu-NOTA-boronsome in 4T1 bearing mice 12 h post injection. **f** Tumour-to normal tissues ratios (T/N) ($n = 4$) by biodistribution at 12 h. Data are means ± SEM. Source data are provided as a Source Data file.

(Supplementary Fig. 4a). As shown in Supplementary Fig. 4b, cell viability was not affected even at the highest dose (40 kBq, $2 \times 10^4$ cells) assessed with a CCK-8 assay, which proved excellent in vitro safety profile for radioactive-labelled boronsomes. Then radiolabelled boronsomes (7.4 MBq, 150 μL) were injected intravenously into a female 4T1 tumour-bearing BALB/c mouse for 6–8 weeks, which showed no obvious systemic toxicity from the results of the serum biochemical test, routine blood analysis and HE staining (Supplementary Fig. 4c, 5). PET studies were conducted to evaluate the tumour specificity and biodistribution of boronsome in vivo.

As shown in Fig. 4a, the only upper part of the mouse body was exposed to thermal neutrons during irradiation, including the tumour and head, forelimbs, and thorax, while the abdomen and hindlimbs were well-shielded with limited exposure. Therefore, biodistribution and accumulation of boronsome within the irradiation radius are of particular concern. Through PET studies, we discovered that [$^{64}$Cu]Cu-NOTA-boronsome accumulated in 4T1 xenograft to give a high tumour-to-normal contrast at 12 hours after injection; meanwhile, boronsomes passed through the central circulatory system after 8 h, then to its retention in the tumour for at least 40 h (Fig. 4b). On the contrary, the uptakes of lungs, muscle, brain, fat, bone, heart and blood were significantly low. Relatively high biliary excretion led to certain liver and intestine uptake, which was within expectations. The rest of radioactivity was excreted through the renal with low kidney retention (Supplementary Fig. 4d). Boron concentrations of

certain organs were measured by ICP-OES, and the average tumourous boron concentration was 93.3 ppm 12 h post injection and 98.9 ppm 8 h post injection (Fig. 4c, Supplementary Table 5, Supplementary Fig. 6), which have met the clinical requirement, but the T/N ratios of each organ at 12 h was better than that at 8 hours (Fig. 4d, Supplementary Fig. 7). By taking both relative and absolute values of boron distribution into consideration, we decided to set the neutron irradiation time as 12 h post injection. Biodistribution study at 12 h showed that 4T1 tumour uptake was $5.49 \pm 0.22$ %ID/g, while uptakes of surrounded normal tissues including breast cancer related high-risk tissues displayed low uptake which also happened to be located in the irradiation area thus gave out a clear background, corroborating the PET-imaging results (Fig. 4e). Moreover, the tumour-to normal tissue (T/N) ratios were investigated (Fig. 4f). The ratios of tumour-to muscle, fat and bone were $36.9 \pm 1.4$, $26.6 \pm 1.5$ and $16.9 \pm 1.1$, respectively, indicating the high tumour selectivity of boronsome in the mouse model. Correspondingly, the boron concentration in the tumour peaked at $40.03 \pm 1.5$ ppm at 0.5 h after injection of 500 mg/kg of BPA with a T/N ratio of 1.83, and $30.10 \pm 1.8$ ppm at 1 h after injection with a tumour-to blood ratio is 2.77 (Supplementary Fig. 8). Therefore, boronsome has advantages over BPA in boron enrichment in tumour and tumour-to blood ratio at tumour sites.

To further evaluate the anticancer effect and safety of boronsome in vivo, we established subcutaneous 4T1 breast tumour-bearing BALB/c mouse model. Female mice were treated by

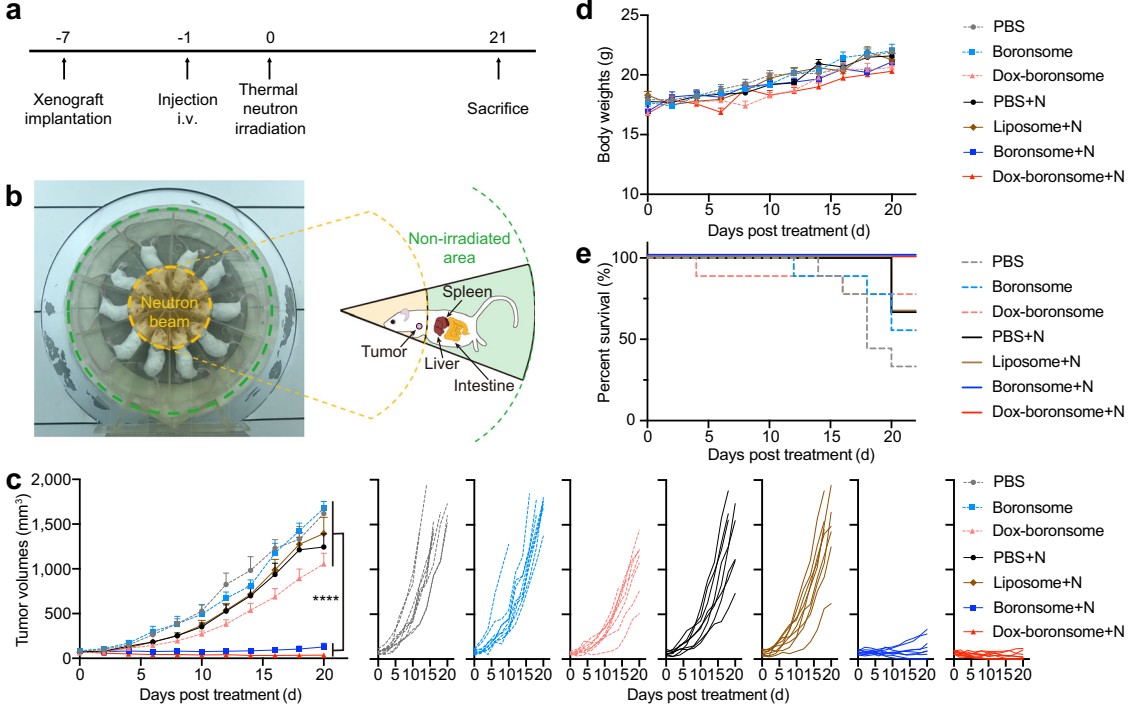

**Fig. 5 Chemotherapy-assisted BNCT provided excellent suppression of tumour growth. a** Experimental flow chart of treatment. **b** Photograph of thermal neutron beam and special-made mold at IHNI. Relative positions of main organs and tumours were shown in the sketch. **c–e** Female 4T1 tumour-bearing BALB/c mice of 6–8 weeks were intravenously injected with PBS, liposome, boronsome or Dox-boronsome, several groups were irradiated with thermal neutrons (+N). **c** Average tumour volumes ($n = 9$) of each group of mice (One-way ANOVA test with Tukey's multiple comparisons test for data of 20 days post treatment, ****$p < 0.001$). **d** Average body weights ($n = 9$) of each group of mice. **e** Survival fractions ($n = 9$) of each group of mice. Source data are provided as a Source Data file.

intravenously injection with phosphate-buffered saline (PBS, 400 μL), boronsome (500 mg/kg), or Dox-boronsome (500 mg/kg) 12 h before neutron irradiation, followed by irradiation with a thermal neutron beam under a neutron flux of $1.9 \times 10^9$ neutrons cm$^{-2}$ s$^{-1}$ for 30 min, then were sacrificed at day 21 (Fig. 5a). During irradiation, mice were anaesthetised and placed lying sideways evenly in a specially-made circular pattern with their heads pointing toward the centre of the neutron beam. Tumours were exposed to thermal neutrons, while livers, spleens and intestines were in a non-irradiated area avoiding damage (Fig. 5b). Dosimetric evaluation of irradiation including γ-photons, $^{14}$N(n,p)$^{14}$C and $^{10}$B(n,α)$^7$Li reactions were based on mean boron concentrations of tumour and normal tissues 12 h post administration. The total dose of the cancerous tissue was 1.94 Gy which was acquired from SERA simulation (Supplementary Fig. 9, Supplementary Table 6).

Tumour volumes measurement was performed routinely to assess treatment effectiveness. For the group of mice treated by PBS, boronsome, PBS+N (+N means with neutron irradiation) and liposome+N, the volume of tumours increased by approximately 20-fold in three weeks. For the group treated with Dox-boronsome, the tumour volumes increased by 14.5-fold, showing partial yet insufficient treatment efficacy. In contrast, upon the treatment with boronsome+N and Dox-boronsome+N, remarkable suppressions on tumour growth have been observed, 1.6-fold and 0.5-fold respectively, indicating that treatment combined boronsome (or Dox-boronsome) with thermal neutron irradiation specifically impaired the growth of tumours. Significantly, four of nine tumours treated by Dox-boronsome+N were eradicated within three weeks (Fig. 5c). Besides, the tumour volumes increased by 3.8-fold for the group treated with BPA, showing partial yet insufficient treatment efficacy compared to

the group treated with boronsomes (0.5-fold) after 14 days (Supplementary Fig. 10a). Additionally, tumour-bearing mice receiving boronsome+N and Dox-boronsome+N treatment had a 100% survival rate which was higher than that of other groups of mice (Fig. 5e), further validating the therapeutic effect of boronsome-BNCT and the synergistic effect of combined treatment with boronsome-BNCT and chemotherapy.

None of the mice showed any apparent deleterious effects from boronsome-BNCT treatment. No radiation or biochemical toxicity was observed since boronsome selectively accumulated only in tumours within the irradiation area, meanwhile, thermal neutron and boronsome alone exhibited minimal toxicity to normal tissues. Although significant boron accumulation was observed in abdominal organs (including intestine liver and spleen), these organs were outside of the irradiation area and were free from any noticeable damage. In addition, our data showed that the body weights of mice did not show any decreases for all treatment groups (Fig. 5d), and no tissue damage was observed in major organs (heart, liver, spleen, lungs and kidneys) according to histology analysis (Supplementary Fig. 5) with no described different changes in routine blood and blood biochemical parameters (Supplementary Fig. 11). Collectively, these results demonstrated that chemo-assisted BNCT with boronsome represented a powerful antitumour strategy with good biological safety.

Exposure to high-LET particles from BNCT leads to DNA damage and poses serious challenges to DNA repair systems. Therefore, we wondered whether inhibiting the repair of DNA damages can be a determinant to improve BNCT efficacy. Poly (ADP-ribose) polymerase-1 (PARP1) is a major enzyme in the repair of DNA strand breaks, and it has been well established that inhibition of PARP1 increases the efficacy of radiotherapy[32–34].

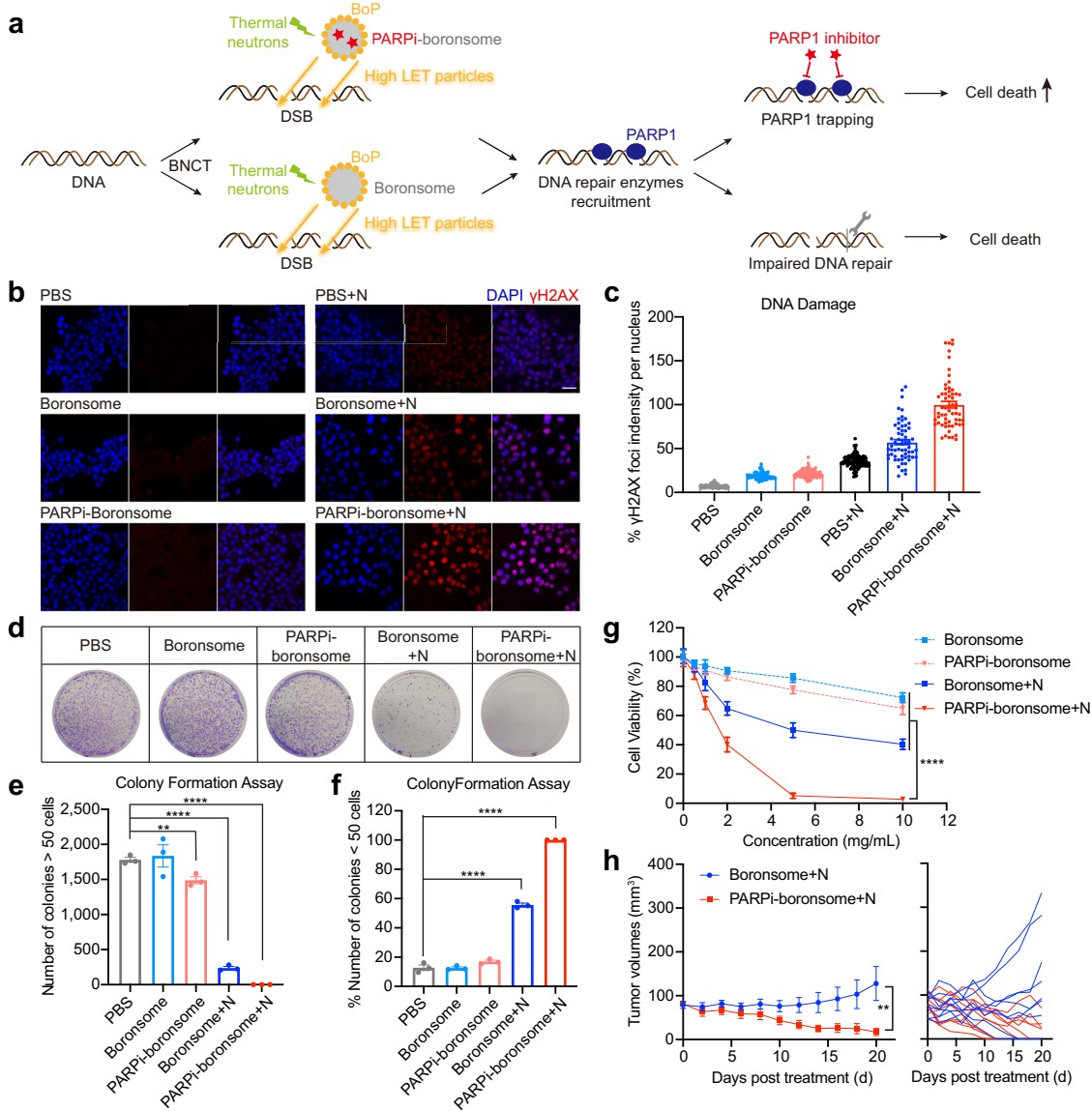

**Fig. 6 DNA damages repair was significantly inhibited by BNCT with PARP1 inhibitor–loaded boronsome. a** Schematic illustration of the synergy between PARP1 inhibition and BNCT. **b** Representative immunofluorescence images were recorded using confocal microscopy. Nuclei were stained with DAPI (blue) and antibody to γH2AX (red). Scale bar, 50 μm. **c** Quantification relative γH2AX red fluorescence in densities per nucleus. Each dots represents the data in a nucleus shown in **b**. PBS, 124 cells; Boronsome, 79 cells; PARPi-boronsome, 167 cells; PBS + N, 117 cells; Boronsome + N, 61 cells; PARPi-boronsome + N, 60 cells. Three independent experiments were performed and representative results are shown. **d**–**f** Colony formation assay of 4T1 cells cultured in 10-cm dishes treated with boronsome, PARPi-boronsome, boronsome+neutron (N) and PARPi-boronsome+neutron (N). **d** Colonies were stained with crystal violet. **e** Quantification of the number of colonies consisting of at least 50 cells ($n = 3$, two-tailed unpaired Student's $t$ test, **$p = 0.0087$, ****$p < 0.0001$). **f** Percentage of the number of colonies <50 cells ($n = 3$, two-tailed unpaired Student's $t$ test, ****$p < 0.0001$). **g** Cell viability of 4T1 cells under various conditions, detected by CCK-8 assay 24 h post treatment ($n = 6$, two-tailed unpaired Student's $t$ test for data of 10 mg/kg, ****$p < 0.0001$). **h** Average tumour volumes ($n = 9$) of each group of mice (two-tailed paired Student's $t$ test, ** $p = 0.0025$). Source data are provided as a Source Data file.

Olaparib is the first PARP inhibitor that has been approved by FDA for gBRCAm metastatic breast cancer, we hypothesised that the combination of Olaparib and BNCT may have a stronger synergistic antitumour effect. Notably, since the DNA repair process initiates immediately after DNA damage, the simultaneous delivery of Olaparib is essential to maximise inhibition of DNA repair. Herein, boronsome-encapsulated Olaparib was developed to validate the therapeutic outcome of combination therapy with PARP inhibitor (PARPi) and BNCT (Fig. 6a).

To quantify DNA double-strand breaks (DSB) induced by boronsome-BNCT and the enhanced efficacy due to the persistence of DNA damage driven by PARP1 inhibition, we performed γ-H2AX staining in 4T1 cells 2 hours after irradiation, which has been adopted as a robust technique to quantify unrepaired DNA DSBs. We found DSB foci increased within cells in the presence of boronsome+N, compared to the groups of boronsome alone and PBS + N (Fig. 6b, c). Remarkably, after neutron irradiation, DSBs induced by PARPi-boronsome were 1.75 times that of

normal boronsome, indicating an evident disruption of DNA repair process.

Additionally, a colony formation assay was performed to evaluate cell proliferation based on the ability of a single cell to grow into a colony through clonal expansion[35], the colonies were fixed at 7 days after irradiation and stained with crystal violet (Fig. 6d). We discovered that treatment with boronsome+N immensely reduced the number of colonies consisting of at least 50 cells compared to groups of PBS, boronsome and PARPi-boronsome, with an increasing percentage of colonies contained <50 cells (Fig. 6e, f). In accordance with expectation, no visible colony was observed in the group treated by PARPi-boronsome +N, which further proved that cell proliferation ability was greatly compromised due to inhibition of DNA repair after concurrent BNCT. These findings were consistent with cell viability measured with the CCK-8 assay, which displayed that all groups showed a concentration-dependent cytotoxicity behaviour (Fig. 6g). At the concentration of 5 mg/mL, cell viability was down to 5.16% in the group of PARPi-boronsome+N, 9 times lower than that of boronsome+N. Overall, the in vitro results illustrated that PARPi amplifies the efficacy of BNCT in a DNA repair interference manner, promoting cell death mode switching from reproductive death to functional death.

The average volume of tumours in female 4T1 bearing BALB/c mice treated with boronsome+N increased by 1.6-fold in 3 weeks, while average tumour volumes of PARPi-boronsome+N group shrank to one-fifth of the average original volume (Fig. 6h), demonstrated a better suppressive effect than Dox-boronsome-BNCT and other control groups (Supplementary Fig. 10b, c). No significant toxicity was found during the experiments (Supplementary Fig. 10d). Notably, five of nine tumours treated by PARPi-boronsome+N were eradicated within three weeks (red), comparing to two of nine in boronsome+N group (blue). Therefore, after thermal neutron irradiation, PARPi-boronsome significantly reduced tumour burden than boronsome. Taken together, these data demonstrated the synergistic effect of BNCT generating substantial DNA damages and PARP1 inhibitors interfering DNA repair.

In this work, we developed boronsome as a boron-enriched lipid bilayer for combinational BNCT. Molecular dynamics simulation and characterisation indicated that carborane-contained phospholipids can form biomimetic nanovesicles with high stability. Meanwhile, boronsome can transport sufficient boron into tumour cells selectively without exhibiting toxicity. Moreover, simultaneously delivery of PARP inhibitors with boronsome demonstrated the synergy between DNA damages and DNA repair system interference. We expected that such a concurrent chemoradiotherapy strategy can be applied in clinical BNCT practice to reduce costs and improve therapeutic outcomes.

## Methods

All animal experiments were performed according to the Animal Protection Guidelines of Peking University, China. All animal care and experimental procedure were performed by following the animal protocols (CCME-LiuZB-2) approved by the ethics committee of Peking University.

**Chemical synthesis of thiocarboranyl triethylamine.** The substrate (o-carborane, 1.00 g, 6.93 mmol) was dissolved in THF (30 mL) and cooled at 0 °C in dry and anaerobic conditions. Then n-BuLi (4.4 mL, 6.93 mmol) was added to the solution dropwise for 15 min, and stirred for 30 min. Then the solution was warmed up to r.t. for another 30 min. Then sulfur(α) (225 mg, 6.93 mmol) was dissolved in the solution above at 0 °C to react for 30 min. Then the solution was warmed up to r.t. for another 30 min. The solution was then evaporated in vacuo to produce a yellow liquid. Then HCl (1 M, 7 mL) was added to quench the remaining n-BuLi. The solution was extracted with hexane and washed with water and brine, then evaporated in vacuo. In a solution of 1-mercapto-1,2-dicarba-*closo*-dodecaborane in hexane (6 mL) at room temperature, Et₃N (1 mL, 7.18 mmol) was added dropwise

and stirred for 15 min. The white precipitate formed was filtered, washed with 3 × 6 mL of hexane and dried in air (compound 3, 1.26 g, 66% yield).

**Chemical synthesis of carboranyl fatty acids.** Synthesis of 1-C₂H₅OC(O) (CH₂)ₙBr (compound **8–11**) (n = 3, 7, 11, 15)

To a solution of 1-HOOC(CH₂)ₙBr (9.3 mmol) in ethanol (50 mL), 5 mL of concentrated sulfuric acid was added dropwise, stirred for 15 min at room temperature and heated under reflux for 12 h. The reaction mixture was cooled and evaporated to dryness in vacuo. The residue was treated with ethyl acetate (30 mL) and water (30 mL). The organic layer was separated, washed with water and saturated brine, dried over Na₂SO₄ and evaporated in vacuo. The crude product was purified using column chromatography on silica with petroleum ether and ethyl acetate as eluent. The solvent was evaporated under a vacuum to yield a yellow oil.

Synthesis of 1-C₂H₅OC(O)(CH₂)ₙS-1,2-C₂B₁₀H₁₁ (compound **12–15**) (n = 3, 7, 11, 15)

To a solution of 1 (1.86 mmol) in ethanol (42 mL), Br(CH₂)ₙC(O)OC₂H₅ (1.86 mmol) was added, stirred for 15 min at room temperature and heated under reflux for 16 h. The reaction mixture was cooled and evaporated to dryness in vacuo. The residue was treated with ethyl acetate (30 mL) and water (30 mL). The organic layer was separated, washed with water and saturated brine, dried over Na₂SO₄ and evaporated in vacuo. The crude product was purified using column chromatography on silica with petroleum ether and ethyl acetate as eluent. The solvent was evaporated under a vacuum to yield a yellow oil.

Synthesis of 1-HOOCC(O)(CH₂)ₙS-1,2-C₂B₁₀H₁₁ (compound **16–19**) (n = 3, 7, 11, 15)

1-C₂H₅OC(O)(CH₂)ₙS-1,2-C₂B₁₀H₁₁ was dissolved in glacial acetic acid (29 mL) forming a clear colourless solution. After stirring for 15 min at room temperature, water (9.8 mL) and concentrated sulfuric acid (9.8 mL) were added. The reaction mixture was heated at reflux for 16 h. The reaction mixture was cooled and poured into cold water (30 mL). The white precipitate was extracted with petroleum ether. The organic phase was collected and evaporated *in vacuo* to yield products as white solid[36,37].

**Chemical synthesis of BoPs.** To a solution of 0.1 M carboranyl fatty acid (2.0 eq.) in chloroform was added N,N-dicyclohexylcarbodiimide (2.6 eq.) and N,N-dimethyl-4-aminopyridine (4.0 eq.). The solution was stirred at r.t. for 5 min. Lyso-PPC (1.3 eq.) was then added and the reaction was stirred for 24 h. The reaction mixture was concentrated using rotary evaporation. Purification by silica flash chromatography (65:25:4 CH₂Cl₂:MeOH:H₂O) yielded BoP species (compound **21–24**)[38].

**Boronsome preparation.** Boronsomes were prepared by the same method as previously described[39]. To generate 1 mL of boronsomes (total lipids 4 mg/mL) of the indicated formulations (BoPs: DPPC: Cholesterol: DSPE-PEG2k = 50:10:40:1, mol%), lipids were dissolved in 0.2 mL ethanol at 60 °C, followed by injection of 0.8 mL of 250 mM ammonium sulfate (pH 5.5) buffer at 60 °C. The boronsome solutions were then passed 10 times at 60 °C through sequentially stacked polycarbonate membranes of 0.2 and 0.1 μm pore size using an extruder (Avanti Polar Lipids). Free ammonium sulfate was removed by dialysis in a 500 mL solution composed of 10% sucrose and 10 mM histidine (pH 6.5) with at least three buffer exchanges.

**Cargo loading.** For SRB loaded boronsomes, lipids of the indicated formulations were dissolved in ethanol and hydrated with 50 mM SRB, sonicated at 45 °C for 30 min. Doxorubicin (Dox) or Olaparib (PARPi) was loaded via the ammonium the ammonium sulfate gradient method. Dox or Olaparib with a drug to lipid molar ratio of 1:8 was added into the liposome solution and incubated at 60 °C for 1 h. Stability and encapsulation ability were evaluated by SRB release which was measured by absorbance using the formula: %SRB release = $(A_{final} - A_{initial})/(A_{Triton-X-100} - A_{initial}) \times 100\%$.

**Characterisation.** Intensity-average hydrodynamic diameters and size distributions (PDI) were measured by Dynamic laser light scattering (DLS) which were conducted on a Zetasizer Nano ZS system (Malvern Instruments Ltd, England). The scattered light was collected at a fixed angle of 173° for duration of ~5 min. All data were averaged over three consecutive measurements. The morphological examination of boronsome was performed with TEM operated at 200 kV (JEM-2100, JEOL, Japan). A drop of boronsome solution was added to a copper mesh covered with an ultrathin carbon film and dried at room temperature for 10 min. The excess solution was removed before negatively staining with uranyl acetate solution (4% w/v) followed by washing with water.

**Preparation of [⁶⁴Cu]Cu-NOTA-boronsome.** ⁶⁴CuCl₂ in 0.01 M hydrochloric acid was provided from Peking University Cancer Hospital. NOTA-boronsome was prepared as described at the indicated formulation (BoP-3: DPPC: Cholesterol: DSPE-PEG2k-NOTA = 50:10:40:1, mol%) before use. Briefly, boronsome solution (0.5 mg/mL) was made up to 1 mL of NaAc buffer (pH 5.5, 0.2 M), and incubated with ⁶⁴CuCl₂ (185 MBq) at 37 °C for 2 h[40]. [⁶⁴Cu]Cu -NOTA-boronsome was

purified through PD-10 chromatography with PBS as an eluent. Radiochemical yield was up to 90.3%.

**Radio-TLC.** Stability of [$^{64}$Cu]Cu-NOTA-boronsome was determined by thin-layer chromatography with the eluant as EDTA solution. Radio-signal of [$^{64}$Cu]Cu-NOTA-boronsome on paper stripes at various time points were collected by Scan-RAM (LabLogic).

**3D structures modelling.** Spartan 14 software was used to predict the 3D structure of BoPs after optimised at Density Functional Theory/Becke, 3-parameter, Lee-Yang-Parr, and 6-31 G* basis set (DFT/B3LYP/6-31 G*). Parameters including $I_c$, carbon arm length (Å); $I_b$, boron arm length (Å); $\Delta l$, length difference between lipid arms (Å); $d$, the distance between lipid arms (Å) and $\theta$, the dihedral angle between lipid arms (°) were measured and evaluated.

**MD simulation.** We simulated the dynamics trajectory of lipid bilayers systems consisting of 128 lipid molecules (BoP-1, BoP-2, BoP-3, BoP-4 and DPPC). For BoP-n lipid bilayers systems which are composed of 108 BoP-n molecules, 20 DPPC molecules and 3200 water molecules. The dynamics simulations were carried out in an NPT ensemble (298.15 K, 1 atm), and we took 10 ns simulations per systems.

The Molecular Dynamics simulations were performed by GROMACS 2018 software[41]. We used Automated force field Topology Builder (ATB, 3.0) tool[42] to generate the force field parameter and topology information. The GROMOS 54a7 force field modified by ATB[43,44] ware utilised to describe this system. VMD 1.9.3 was used in the computational study for visualisation.

We used the chain order parameter (Szz) to estimate the degree of regularity of hydrocarbon chain. It was defined as

$$S_{ZZ} = \frac{3 < \cos \theta_i >^2 - 1}{2}$$, $\theta$i represents the angle between bilayer normal and the vector of Ci-1 to Ci+1. We chose $z$ axis as bilayer normal and Szz indicates the orientation of hydrocarbon chain with respect to the bilayer normal. When Szz is equal to 1, the hydrocarbon chain parallel to bilayer normal. We calculated the average chain order parameter of hydrocarbon chain in 5 to 10 ns (See Fig. 1f). Detailed information about the parameters when performing the MD simulation was listed (Supplementary Tables 1–3).

**Irradiation of cells and animals.** Cells or animals were irradiated at In-Hospital Neutron Irradiator (IHNI) based on miniature neutron source reactor. For cell irradiation, 4T1 cells (3101MOUSCSP5056, obtained from National Infrastructure of Cell Line Resource) cultured in 96-well plates were incubated with boronsome (5 mg/mL) or PRAPi-boronsome (same boron dose) for 24 hours, and were irradiated for 10 min. For animal irradiation, thermal neutron irradiation experiments were performed when the tumour volume of 4T1 tumour-bearing mice reached about 70 mm³. Each mouse in the experimental group was intravenously injected with liposome (500 mg/kg), boronsome (500 mg/kg), Dox-boronsome (same boron dose) or PRAPi-boronsome (same boron dose) while each mouse in the control group was intravenously injected with PBS. After 12 hours, mice anaesthetised with isoflurane (5%, 100 μL) were fixed onto the special-made module (Fig. 5b), and were irradiated for 30 min.

**Determination of minimal boron requirement.** The neutron capture cross-sections ($\sigma$) vary significantly between nuclides. Because the thermal neutrons are preferentially captured by $^{10}$B nuclei (3837 barn, capture cross-section many times greater than that of C (0.0037 barn), H (0.332 barn), O (0.0002 barn), and N (1.75 barn) of which tissues are mainly composed of), and cell damage is mainly caused by high-LET particles generated from $^{10}$B(n,α)$^{7}$Li reaction, concentrations of $^{10}$B in targeted tissues sufficient to produce therapeutic doses is of vital for BNCT treatment.

$$\text{Neutron capture section}\,(\sigma) = \frac{\text{Capture reaction number per unit time}\,(n_r)}{\text{Incident neutron number per unit time}\,(I) \times \text{Nuclide number per unit area}\,(n_i)} \quad (1)$$

$$\text{Neutron capture cross section of 10B}\,\sigma = \frac{\text{Capture reaction number}\,(N_{reaction})}{\text{Neutron flux}\,(\phi) \times \text{Irradiation time}\,(T) \times \text{10B number}\,(N_B)} \quad (2)$$

Neutron flux ($\phi$) of IHNI-1 = $1.9 \times 10^9$ cm$^{-2}$ s$^{-1}$
Irradiation time ($T$) = 10 min (cells) or 30 min (animals)
To ensure at least one successful $^{10}$B(n,α)$^{7}$Li reaction within one cell, $N_{reaction} \geq 1$, $10^9$ of $^{10}$B ($N_B$) is usually considered to be the minimum requirement.

**Dosimetry by SERA.** The magnitude of the radiation dose of BNCT is complicated due to the radiation field in BNCT consists of several separate radiation dose components including B-10 dose (from $^{10}$B(n,α)$^{7}$Li reaction), gamma dose (from hydrogen $^{1}$H(n,γ) in tissue), N-14 dose (from $^{14}$N(n,p)$^{14}$C in tissue) and hydrogen dose (from $^{1}$H(n,n')p in tissue) with different physical properties and biological effectiveness. The Monte Carlo computation-based dose calculation was performed with SERA (Simulation Environment for Radiotherapy Applications; Idaho National Laboratory, Idaho Falls, ID).

**Cell culture method.** The 4T1 cell lines (3101MOUSCSP5056) were obtained from National Infrastructure of Cell Line Resource (Beijing, China). Cells were cultured in RPMI 1640 medium supplemented with 10% fetal bovine serum and 1% penicillin/streptomycin. All the cells were cultured in a humidified 5% CO$_2$ incubator at 37 ˚C and the medium was replaced every 1–2 days.

**Cell viability assays.** 4T1 cells were seeded in 96-well plates at a concentration of $5 \times 10^3$ cells per well and incubated overnight in a cell culture incubator. Up to 400 kBq/mL [$^{64}$Cu]Cu-NOTA-boronsome, boronsome (5 mg/mL) and PARPi-boronsome (same boron dose) were added to the wells (100 μL final volume per well) with different concentrations. After 24 h incubation at 37 °C, cells were irradiated by thermal neutron beam. And after 24 h post irradiation, cell viability was assessed with the CCK-8, following the manufacturer's protocol.

**Colony formation assay.** To determine cell reproductive death after treatment, 10,000 cells are seeded out in a 10-cm culture dish to form colonies in 7 days. Colonies are fixed with methanol and stained with crystal violet (0.02% w/v).

**Cell uptake studies.** Cell uptake studies were performed on 4T1 cells. Briefly, 4T1 cells ($1 \times 10^5$ per well) were seeded in 24-well plates and incubated overnight at 37 °C. A series of concentration gradient solutions with 0.1–10 mg/mL boronsome (Fig. 3g), boronsome (5 mg/mL), FBY (same boron dose) and BPA (same boron dose) (Fig. 3h) were added to wells, and the mixture was incubated at 37 °C for 24 h. The cells were then carefully washed three times with ice-cold PBS and lysed with 0.1 M HNO$_3$. We determined the cellular boron concentration by ICP-OES: the crudely treated cells obtained in the previous step were digested using a microwave accelerated reaction system (Mars; CEM), and then diluted with deionized water. The boron concentration was determined by ICP-OES on a PerkinElmer Optima 7000 DV according to the published method.

**γ-H2AX Immunofluorescence staining.** The cells were fixed in 4% paraformaldehyde for 15 min, permeabilized with 0.3% Triton X 100 for 20 min and blocked with 10% goat serum for 30 min at room temperature (RT). After blocking, the cells were incubated with anti-γH2AX antibody (1:200, abcam, ab81299) at 4 °C overnight and then incubated with the secondary antibodies, goat anti-rabbit IgG H&L (Alexa Fluor® 488, 1:1000, abcam, ab150077) for 1 h at room temperature in the dark. The cells were stained in 4′,6-diamidino-2-phenylindole (DAPI, Santa Cruz Biotechnology) for 15 min at RT, covered with glycerol-PBS (1:1) and coverslips, sealed with clear nail polish and examined under a laser scanning confocal microscope (Nikon, Japan). Quantification of fluorescence intensities was analysed by ImageJ 1.53a.

**Animal studies.** In all, 6–8-weeks female BALB/c mice were obtained from Beijing Vital River Laboratory Animal Technology Co, Ltd. and maintained under specific pathogen-free facility conditions with a 12 light/12 dark cycle, and free access to food and water. Mice were housed under temperature of 24 ± 2 °C, and humidity of 50 ± 10%.

**Xenografts implantation.** Subcutaneous 4T1 tumour model was established by injection of $10^6$ 4T1 cells in 100 μL PBS into the right flank of female BALB/c mice. After 1 week, all drugs were administered to mice by intravenous (i.v.) injection 12 hours before neutron irradiation.

**Therapeutic evaluation.** 4T1 tumour diameters were measured by caliper, and the tumour volume ($V_t$, mm³) was calculated by $V_t = 0.5 \times a \times b^2$, where $a$ was the long axis and $b$ was the short axis. Body weight was recorded every 2 days. Data were collected and the relationship between tumour volume and time was plotted, the therapeutic effects of different groups were evaluated from the experimental results. When the tumour size reached 2000 mm³ or the loss was >20% of total body weight, the mice were removed from the experimental group and euthanized.

**Blood routine and biochemical test.** Mice bearing 4T1 tumour were injected with 5 mg/kg of [$^{64}$Cu]Cu-NOTA-boronsome intravenously (7.4 MBq) or 500 mg/kg boronsome, and then blood routine and serum biochemical were analysed according to the standard method at different time.

**Small-animal PET/CT imaging.** Mice bearing 4T1 tumour were injected with 5 mg/kg of [$^{64}$Cu]Cu-NOTA-boronsome intravenously (7.4 MBq) and anaesthetised by inhalation of 2% isoflurane/oxygen mixture and placed on a scanner bed approximately 10 minutes prior to PET/CT image acquisition. PET scans were performed on Nanoscan PET-CT 122 s (Mediso Medical Imaging Systems). PET acquisitions for static reconstructions were performed 0.5 h, 2 h, 4 h, 8 h, 12 h, 24 h and 48 h after injection. For each acquisition, 3D regions of interest (ROI) were

acquired by Nucline NanoScan software (InterViewTM FUSION, Mediso Medical Imaging Systems) on the decay-corrected images. The radioactivity in relevant tissues was obtained from mean and maximum standard uptake values (SUVs) of the ROIs and then converted to the percentage injected dose per gram (% ID/g, assuming a tissue density of 1 g/mL).

**Pharmacokinetics study.** We determined the cellular boron concentration by ICP-OES: mice that were injected intravenously with 500 mg/kg boronsome or BPA were sacrificed at different time points, then obtained tumour, blood, heart, liver, spleen, lung, muscle. kidney and brain. These organs were lysed with $HNO_3$ and were digested using a microwave accelerated reaction system (Mars; CEM), then diluted with deionized water. The boron concentration was determined by ICP-OES on a PerkinElmer Optima 7000 DV according to the published method.

**H&E staining.** The major organs including the heart, liver, spleen, lung and kidney were excised from mice of different groups 21 days after treatment for the histopathologic study. Organs were fixed in 4% paraformaldehyde, embedded with paraffin, sectioned into slices, and stained with hematoxylin and eosin. Samples were chosen at random, and the slices were photographed by 3Dhistech.

**Statistics analysis.** Statistical analyses were performed using GraphPad Prism 8. All data were expressed as mean ± SEM.

**Reporting summary.** Further information on research design is available in the Nature Research Reporting Summary linked to this article.

## Data availability
All data supporting the findings of this study are included in the Article and its Supplementary Information. Source data are provided in this paper.

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

## Acknowledgements

This study was funded by the National Nature Science Foundation of China (Grant no. U1867209), the Ministry of Science and Technology of the People's Republic of China (Grant nos. 2021YFA1601400 and 2017YFA0506300), the Beijing Municipal Natural Science Foundation (Grant no. Z200018), the Special Foundation of Beijing Municipal Education Commission (Grant No. 3500-12020123), Li Ge-Zhao Ning Life Science Youth Research Foundation (LGZNQN202004) to Z.L. We thank the facility support from the Analytical Instrumentation Centre of Peking University.

## Author contributions

Z.L. conceived the study. J.L. assisted by Q.S. C.L. Z.G. and D.D. performed material synthesis, characterisation and chemical analysis. J.L. assisted by Q.S. C.L. and Z.G. performed radiosynthesis, PET imaging and data analysis. Q.S. assisted by J.L., C.L. and Z.G., performed cell studies and animal studies. H.X. performed theoretical analysis. T.L. and Z.Z. provided thermal neutron source and performed SERA simulation. J.L. assisted by Q.S., C.L., Z.G. and D.D. performed all other experiments. J.L. and Q.S. analysed the data. J.L., Q.S. and Z.L. wrote the manuscript with inputs from all authors. All authors discussed the results and commented on the manuscript.

## Competing interests

The authors declare no competing financial interests.
