## [Peer Review File · Nature Communications]

REVIEWER COMMENTS

Reviewer #1 (Remarks to the Author): Expert in BNCT and nanodelivery

This manuscript submitted by Li et al. describes the development of the novel boron lipid liposomes. In liposomal boron delivery system, two types of liposomes have been reported: boron-encapsulated liposomes (Scheme 1a) and boron-lipid liposomes (Scheme 1b). The latter strategy has used boron ion clusters as a hydrophilic moiety in the lipid. In this study, the authors demonstrated an alternative approach using a lipophilic function of carborane. The approach is interesting, and a variety of evaluation methods have been implemented to demonstrate the properties of this liposome called "borosome". However, this reviewer wonders how advantageous the current borosomes are compared to the reported boronated liposomes or even to BPA. There are no controls in the experiments demonstrated in this paper. In addition, the PET imaging demonstrated in Fig. 3b shows long-term retention of ^{64}Cu -NOTA-Borosome in tumor tissue. To confirm the safety issue of borosome, the authors need to verify the pharmacokinetics of it for a long time, especially whether it will be excreted. In this regard, the time-dependent boron concentration in each organ is essential to determine the compound biological effectiveness (CBE) factors that are necessary for the use of SERA system.

Furthermore, MD simulations have been performed to obtain molecular properties of lipid bilayers. However, this reviewer is wondering what parameters were applied to the carborane clusters to perform the MD simulation. The detailed information about the parameters should be provided. In conclusion, the studies presented in this paper look brilliant at first glance but lack several elements essential for the evaluation of boron agents in BNCT. Therefore, this reviewer is hesitant to recommend this manuscript for Nature Communications, which has a significant impact on society.

Reviewer #2 (Remarks to the Author): Expert in BNCT and imaging

The paper by Li et al., entitled "Borosome as A Liposome Mimic for Combinational Neutron Capture Therapy", presents the preparation and characterization of boron-rich liposomes as potential BNCT agents. The authors report on the *in vitro* characterization, and subsequent *in vivo* biodistribution in a mouse model of breast cancer using *in vivo* imaging techniques. Therapeutic efficacy of the borosomes alone and in combination with loaded drugs are evaluated in this tumor model, with interesting results when attending to tumor size progression.

The work reported by the authors is extensive and covers from preparation to *in vivo* evaluation. The therapeutic efficacy achieved is promising and hence the paper might be of the interest of the scientific community. However, there are issues that, at the moment, prevent my recommendation for publication.

- The idea of generating the borosomes is interesting. However, the incorporation of boron cages in the lipid bilayer of liposomes has been already reported in the past (middle figure, scheme 1). The authors mention very rapidly the advantages of their system, but maybe they should concentrate more efforts in highlighting the advantages of their nanoformulation when compared to previously reported nanosystems with similar composition.

- It is not clear to the reviewer if the authors use ¹⁰B-rich carborane or not. As the authors know, only 20% of the natural boron is boron-10, and hence this is a critical aspect of the experimental design. The use of ¹⁰B-enriched carborane may lead to a huge cost, although it might be necessary to achieve therapeutic efficacy in the clinics. Indeed, the dose administered by the authors is extremely high (500 mg/Kg). Allometric scaling predicts a dose of ca. 40 mg/Kg in humans, which for a 80 Kg subject would be 3 grams. Is this realistic? The authors should report on the yield of all chemical reactions, to estimate how much carborane will be needed to prepare, e.g., 1 g of boronsomes.
- The authors select 12h as the time point for neutron irradiation. The long residence time in the tumor suggests that later time points could be more appropriate, to decrease Tumor-to-blood ratios. Can the authors provide dynamic time-activity-curves for all organs and the tumor-to-organ ratios? Values at 12 h are provided, but PET images should provide this info at the different time points.
- There are a wide variety of boron-rich nanosystems reported in the literature that achieve >5% of ID/g of tumor, and hence when administered at this dose they would also achieve high concentration in the tumor. The key points here are: 1) which is the percentage of boron (¹⁰B) in the final nanosystem? and 2) how do these values compare to previous nanosystems reported in the literature? These need further consideration in the paper.
- The authors rely in EPR effect to accumulate the boronsomes in the tumor. However, EPR effect is known to be heterogeneous and there is a large inter-subject variability. This deserves a comment mentioning the potential limitations.
- The authors have selected a triple negative breast cancer model for their in vivo experiments, which is not the typical cancer type to evaluate new BNCT agents. As a proof of concept, I do not see any problem. However, TNBC is known to have a bad prognostic because it is likely to be spread at the time it is found. When the cancer is localized, 5-year survival with current therapeutic alternatives is >90%. With distant metastasis, the value drops to 12%. However, with distant metastasis BNCT would not be that useful. This deserves again a comment in the manuscript.
- Characterisation of labelled liposomes and stability: Figure S7 is very poor. Are these TLC profiles? If yes, the X axis is not correct (should be distance). Also, at long times it seems that a second peaks tends to appear on the right... but the curves are cut... the authors need to clarify this.
- For the combination therapies, which is the dose of the drug administered to the animals? It is always 500 mg/Kg in boronsomes + the amount of drug? It is important to clarify which is the amount of boron and drug administered in each case (all groups).
- Citations are quite old most of them. It is true that a huge portion of the work related to BNCT and liposomes was performed a couple of decades ago. However, there are recent works related to boron-rich nanomaterials that have been ignored. Both for citation and for discussion with comparative purposes.
- The TOC file is not representative of the scope of the work and needs improvement

Reviewer #4 (Remarks to the Author): Expert in nanoparticles and therapy

This work presents a clever strategy to prepare liposomes that are able to promote simultaneous delivery of boron and chemotherapy drugs to brain tumors. The innovative strategy stands out for the use of a boron-modified lipid molecule that could integrate the lipid bilayer of a liposome structure and therefore, instead of being leaked from the liposome as usually is found with this type of nanoparticle,

the boron is released from this bilayer. The authors explored several applications and proved that the borosome is useful as drug-delivery platform as well as theranostic platform since it could be also loaded with radioactive chemicals.

In my opinion this work should be considered to be published in Nature Communications but some suggestions should be considered:

- 1) Although the application of borosome as theranostic platform has been shown, in vitro cytotoxicity as well as in vivo toxicity data are missing with the radioactive-labeled borosomes.
- 2) In page 4 it was mentioned the absence of toxicity of radiolabeled borosomes but even if the borosomes were selectively accumulated in the tumor tissue, it was possible to find them in normal tissue but there was no histological studies in these tissues, as it was performed with DOX-loaded borosomes.
- 3) It was evident the lack of recently published research works among the consulted references. The data should be commented comparing the recent results obtained by similar research works. There was only 2 references from 2020 and only 1 from 2021.

Some minor comments:

- 1) In the abstract "unsatisfied boron delivery agents". Please complete the sentence by pointing delivery to "brain tumor" or other suggestion the authors deem appropriated.
- 2) Abstract 4th line: "borosome, a carboranyl-phosphatidylcholine based lipid bilayer for the first time". It would be properly called liposome instead of lipid bilayer because it is not a "sandwich bilayer", it is a spherical nanoparticle as shown in the TEM images.
- 3) Abstract 6th line: "treated by borosome". It should be correct to "treated with".
- 4) First page, last line before scheme 1: "Lack of non-invasive imaging techniques to explore in vivo biodistribution of boron agents. It could be better explained that the problem is not only the missing technique but absence of suitable properties of boron agents to allow their detection by imaging techniques
- 5) First line, page 2: correct "to developed" to "develop".
- 6) First line, page 2, second paragraph: how the stability of borosome was measured? It was implicit that was by dynamic light scattering (DLS) measurement. However, it was not mentioned the technique as well as it was not mentioned the measurement of Hydrodynamic diameter. It is crucial to characterize this property if the mentioned "size" and polydispersity index was not measured by DLS.
- 7) Values of T/N ratio was not appropriately reported since the deviation/error has excessive significant digits. For example, the value 36.9 ± 1.41 should be corrected to 37 ± 1 .
- 8) Page 3, 3rd paragraph: "naive" should be replaced by "unloaded".
- 9) Page 3, 3rd paragraph: It has to be further explained why "electric neutralized" can be considered an evidence of dox loading. In fact, it has to be correctly called "lower Zeta potential". It has to be kept in mind that Zeta potential is an indicative of surface charge and therefore if after DOX loading the Zeta potential was decreased in modulus, this is an evidence that DOX is close to the surface instead of encapsulated.

RESPONSES TO THE COMMENTS

Dear reviewers,

On behalf of the authorship group, I would like to thank the reviewers for taking the time to evaluate our manuscript entitled “**Borosome as a liposome mimic for combinational neutron capture therapy**”. We are deeply grateful for your appreciation on our work. Please kindly find our point-by-point response to reviewers’ comments below.

Reviewer #1

This manuscript submitted by Li et al. describes the development of the novel boron lipid liposomes. In liposomal boron delivery system, two types of liposomes have been reported: boron-encapsulated liposomes (Scheme 1a) and boron-lipid liposomes (Scheme 1b). The latter strategy has used boron ion clusters as a hydrophilic moiety in the lipid. In this study, the authors demonstrated an alternative approach using a lipophilic function of carborane. The approach is interesting, and a variety of evaluation methods have been implemented to demonstrate the properties of this liposome called “borosome”. However, this reviewer wonders how advantageous the current borosomes are compared to the reported boronated liposomes or even to BPA. There are no controls in the experiments demonstrated in this paper.

- We thank the reviewer for his/her enthusiastic comments that our work is novel and interesting. The manuscript and supporting data have been revised accordingly.

As suggested, the advantages of borosome have been evaluated through both literature review and experimental comparisons.

1) Literature review

According to the summary of previously reported boronated liposomes (**Table R1**), we found that several drawbacks have hampered currently-reported boronated liposomes to become practical boron carriers in clinics: a. Liposome structures currently reported are mainly based on boron-enriched small molecules encapsulation strategy, of which the loading capacity is limited and may have cargo leakage to off-tumour tissues¹; b. Application of unusual boranes which are short of *in vivo* stability could induce unexpected biochemical toxicity and immunogenicity²⁻⁴; c. Lack of non-invasive imaging techniques to explore *in vivo* biodistribution of boron agents to ensure the accuracy of neutron irradiation and to improve efficacy⁵.

For more information, the key criteria (tumour selectivity, treatment efficacy, etc.) of previously reported boronated liposomes have been evaluated to further highlight the advantages of borosome (**Table R1**). Of note, the boron content of “borosome” is higher than other reported boronated liposomes and BPA (**Table R2**).

Table R1 Summary of previously reported different boronated liposomes.

No.	Compositions	Boron moiety	Max. boron in tumour (ppm)	Boron Dose (mg/kg mouse)	Tumour/blood ratio	Safety	Efficacy	Drug loading property	Imaging property
1	BoP3, DPPC, CHOL (cholesterol) and DSPE-PEG2000 (50:10:40:1) (This work)	closo -dodecarborane (at lipid tails)	93 (12 h)	39	4.2 (12 h)	No systemic toxicity or side effects have been observed	The tumour volume shrank to 1/5 after about 3 weeks (PARPi-boronsome)	✓ (Dox or PARPi)	✓ (PET-CT)
2 ⁶	DSPC, CHOL, B-lipid, DSPE-PEG-OMe (1:1:0.25:0.11). Transferrin conjugated.	nido -carborane (at lipid heads)	22 (72 h)	7.2	< 4.0 (72 h)	LD ₅₀ (48 h) = 14 mg B/kg	Average survival rate: 21 days (untreated) and 31 days (treated)	×	×
3 ⁷	DSBL, DSPC, CHOL, DSPE-PEG (0.1:0.9:1:0.11)	BSH (at lipid heads and in cavity)	35 (24 h)	15	1.5 (36 h)	At given dose no mouse died within 3 weeks	Tumour controlled within 3 weeks	×	✓ (MRI)
4 ⁸	DSPC, CHOL, and B-lipid (1:1:1)	closo -dodecarborane (at lipid heads)	N.A.	N.A.	N.A.	IC ₅₀ (24 h) = 5.6 mM (V79 Chinese hamster cells)	N.A.	×	×
5 ⁹	DSPC, CHOL, and B-lipid (1:1:1)	closo -dodecarborane (at lipid heads)	N.A.	N.A.	N.A.	IC ₅₀ (24 h) = 2 mM (V79 Chinese hamster cells)	N.A.	×	×

6 ¹⁰	DSPC, CHOL, and MAC (1:1:0.6)	TAC (in cavity) and MAC (in the middle of the bilayer)	27 (30 h)	17 /2 inj:18.6	1.4 (30 h)	No systemic toxicity or side effects have been observed	Tumour volume shrank to 12% of controlled group after 2 weeks	×	×
7 ¹¹	DSPC, CHOL, DSPE-PEG, DSPE-PEG-COOH (2:1:0.11:0.021); Transferrin conjugated	BSH (in cavity)	34 (48~72 h)	35	6.0 (72 h)	No systemic toxicity or side effects have been observed	Tumour controlled within 12 days	×	×
8 ¹²	DSPC, CHOL, B-lipid (1:1:0.6)	nido -carborane (at lipid heads) and BSH (in cavity)	32 (30 h)	11	5.3 (48 h)	No systemic toxicity or side effects have been observed	Tumour controlled within 3 weeks	×	×

Notes: (A) Structure 2, whose boron moiety were *nido*-carborane, had acute cytotoxicity which led 50 % of mice die at a dose of 14 mg B/kg weight within 48 h; (B) Structure 4 and 5 did not perform boron biodistribution or BNCT studies; (C) TAC (Na₃[*ae*-B₂₀H₁₇NH₃]) and MAC (K[*nido*-7-CH₃(CH₂)₁₅-7,8-C₂B₉H₁₁]) were in different positions of the liposomes; (D) The injection dose of boronsome was converted from the value of 500 mg (the mass of boronsome solution)/kg. N.A., not available.

Table R2 Comparison of boron percentage in different boronated liposomes and in BPA.

No.	1	2	3	4	5	6	7	8	BPA
B mass fraction/%	7.9	N.A.	N.A.	5.6	5.6	N.A.	N.A.	N.A.	5.2

Notes: Boron percentage of structure 2, 3, 6, 7 and 8 can't be precisely calculated due to limited information. N.A., not available.

In this work, we aim to develop stable boronated liposomes with good biocompatibility and high boron content. Different from previous reports^{1,13}, the carboranyl group is covalently conjugated to the hydrophobic tail of phospholipids to form a series of boronated phospholipids (BoPs), and a thiol-halo reaction was adopted instead of click chemistry to maintain the flexibility of phospholipids. To achieve good biocompatibility and reduce potential toxicity, structure of phosphocholine was maintained as the hydrophilic head¹⁴. In addition, as the boron-containing moiety to form liposome membranes, the internal cavity is released to carry other drugs for combination therapy (**Figure R1**).

Figure R1. Schematic illustration of boronated liposomes for BNCT.

2) Experimental comparison between BPA and boronsome

As summarized in **Table R1**, boronsome exhibits better tumour treatment efficacy than the previously reported boronated liposomes. We have also performed a comparison experiment between BPA and boronsome to evaluate their BNCT treatment efficacy. BPA (500 mg/kg) fructose complex and boronsome (500 mg/kg) were tail-vein administered to the 4T1 tumour-bearing mice. The boron contents of BPA and boronsome in tumour and major organs have been assessed by ICP-AOES (Inductively Coupled Plasma-Optical Emission Spectrometry), respectively. As shown in **Figure R2**, boronsome shows remarkably prolonged therapeutic window and higher boron uptake in tumours than those of BPA. A high tumour to blood ratio of boronsome was observed during 8 h to 24 h. While the high tumour to blood ratio of BPA could only last for 0.5 h. In addition, the peak tumourous boron concentration of boronsome (98.9 ppm) is significantly higher than that of BPA (40.0 ppm). The following BNCT study corroborates the aforementioned finding in biodistribution study. As shown in **Figure R3**, BPA (500 mg/kg)-BNCT has moderate suppression to tumour growth (the tumour volumes increased by 3.8-fold in 15 days), while boronsome (500 mg/kg)-BNCT exhibits excellent treatment efficacy (tumour volume decreased by 50%). In sum, boronsome provides notably improved treatment efficacy and more favored pharmacokinetics compared to BPA.

Figure R2. Boron biodistribution of BPA and boronsome in 4T1 xenografts bearing mice measured by ICP-OES. a, 0.5, 1, 1.5, 2, 4, 8 h post injection of BPA (500 mg/kg, i.v.). b, 0.5, 2, 4, 8, 12, 24, 36, 48, 72 h post injection of boronsome (500 mg/kg, i.v.). T/B ratio, tumour-to-blood ratio.

Figure R3. Tumour volumes ($n = 4$) of each group of mice treated with Boronsome+N and BPA+N. Paired Student's t-test, $*P < 0.05$, Data are means \pm SEM.

2. In addition, the PET imaging demonstrated in Fig. 3b shows long-term retention of ^{64}Cu -NOTA-Boronsome in tumor tissue. To confirm the safety issue of boronsome, the authors need to verify the pharmacokinetics of it for a long time, especially whether it will be excreted. In this regard, the time-dependent boron concentration in each organ is essential to determine the compound biological effectiveness (CBE) factors that are necessary for the use of SERA system.

- We thank the reviewer for this very constructive advice. As suggested, a time-dependent boron concentration assay has been carefully conducted to investigate the biodistribution and pharmacokinetics of boronsome in tumour-bearing mice. At indicated time point, four tumour-bearing mice were dissected and the organs of interest were processed through microwave digestion, followed by the ICP-OAES assay to determine the boron concentration. As shown in **Figure R2** and **R4**, the tumour uptake of boron would continually increase till reached a plateau (up to 100 ppm) at 8-12 hours post injection. The boron concentration in blood and other major organs gradually declined after the injection, providing high tumour-to-normal tissue ratios during 8-12 hours post injection (**Figure R5**), which was selected as the treatment window for

neutron irradiation. Of note, most of boron would be excreted from the body at 72 hours post injection, which may minimize the concern regarding toxicity or safety issues.

Figure R4. Time-dependent boron concentration in tumour, blood, lung, brain, muscle, fat, bone, heart, liver, and spleen at indicated time points after the intravenous injection of boronsome (500 mg/kg). Data are means \pm SEM (n = 4 for each time point).

Figure R5. Tumour-to-normal tissue ratio (T/N ratio) of boron in tumour, blood, lung, brain, muscle, fat, bone, heart, liver, and spleen at indicated time points after the intravenous injection of boronsome (500 mg/kg). Data are means \pm SEM (n = 4 for each time point).

3. Furthermore, MD simulations have been performed to obtain molecular properties of lipid bilayers. However, this reviewer is wondering what parameters were applied to the carborane clusters to perform the MD simulation. The detailed information about the parameters should be provided.

- As suggested, detailed information about the parameters when performing the MD simulation was listed as below and in supporting information (**Table R3–R5**).

Table R3. No-Bonds parameter

Type	C6	C12
H	0.00000000	0.00000000
HC	8.464e-05	1.5129e-08
B	0.00234062	4.93728e-06
C	0.00234062	4.93728e-06
S	0.0099840064	1.3075456e-05

Table R4. Bonds parameter

Bonds	$b_0(\text{Å})$	kb
H- B_α	1.19	2.6481e+07
H- B_β	1.19	2.6481e+07
H- B_γ	1.18	3.5909e+07
H- B_δ	1.18	3.5909e+07
H-C	1.09	1.2300e+07
S-C	1.67	3.0478e+06
C-C	1.63	8.5653e+05
C- B_α	1.63	8.7100e+06
C- B_β	1.63	4.7200e+06
B_α - B_β	1.78	1.0665e+06
B_α - B_γ	1.76	1.8519e+06
B_β - B_β	1.79	5.6200e+06
B_β - B_γ	1.78	2.6424e+06
B_β - B_δ	1.78	2.7200e+06

$B_\gamma-B_\delta$	1.79	2.1847e+06
$B_\delta-B_\delta$	1.69	1.9257e+06

Table R5. Angles parameter

Angles	θ	Ka
S-C- B_α	114.00	1559.41
S-C- B_β	132.00	760.00
S-C-C	120.00	560.00
C- B_α -C	56.39	2100.95
C- B_α - B_β	124.00	730.00
HC-C- B_α	118.00	7474.41
HC-C- B_β	125.00	375.00
HC-C-C	120.00	505.00
H- B_α -C	120.00	390.00
H- B_α - B_β	132.22	390.00
H- B_β - B_α	119.00	2211.40
H- B_β - B_β	119.74	3398.91
H- B_β - B_γ	121.00	685.00
H- B_β - B_δ	132.00	760.00
H- B_γ - B_δ	121.00	685.00
H- B_γ - B_β	117.00	635.00
H- B_δ - B_γ	121.00	375.00
H- B_δ - B_δ	121.40	690.00
B_α -C- B_α	115.71	1201.92
B_α - B_β - B_β	97.40	469.00
B_α - B_β - B_γ	60.30	531.36
B_α - B_β - B_δ	103.00	420.00
B_β - B_α - B_β	106.75	503.00
B_β - B_β - B_γ	107.57	484.00
B_β - B_β - B_δ	59.90	508.93
B_β - B_γ - B_δ	60.30	531.36
B_β - B_δ - B_γ	59.90	508.93
B_γ - B_δ - B_δ	63.30	2254.27

Reviewer #2

The paper by Li et al., entitled “Boronsome as A Liposome Mimic for Combinational Neutron Capture Therapy”, presents the preparation and characterization of boron-rich liposomes as potential BNCT agents. The authors report on the *in vitro* characterization, and subsequent *in vivo* biodistribution in a mouse model of breast cancer using *in vivo* imaging techniques. Therapeutic efficacy of the boronsomes alone and in combination with loaded drugs are evaluated in this tumor model, with interesting results when attending to tumor size progression.

The work reported by the authors is extensive and covers from preparation to *in vivo* evaluation. The therapeutic efficacy achieved is promising and hence the paper might be of the interest of the scientific community. However, there are issues that, at the moment, prevent my recommendation for publication.

- We thank the reviewer for his/her encouraging comments that our work is interesting and extensive. The manuscript and supporting data have been revised accordingly.

1. The idea of generating the boronsomes is interesting. However, the incorporation of boron cages in the lipid bilayer of liposomes has been already reported in the past (middle Fig., scheme 1). The authors mention very rapidly the advantages of their system, but maybe they should concentrate more efforts in highlighting the advantages of their nanoformulation when compared to previously reported nanosystems with similar composition.

- We thank the reviewer for this constructive advice. As suggested, the potential advantages of boronsome over other boronated lipid bilayers have been summarized as below and added accordingly in the revised manuscript.

In this work, we aim to develop stable boronated liposomes with good biocompatibility and high boron content. According to the summary of previous reports (**Table R6**), we found that the previously reported boronated lipid bilayers often utilize unusual boranes which are short of *in vivo* stability that could induce unexpected biochemical toxicity and immunogenicity²⁻⁴. This drawback may hamper currently-reported boronated liposomes to become practical boron carriers in clinics. In this work, the carboranyl group is covalently conjugated to the hydrophobic tail of phospholipids to form a series of boronated phospholipids (BoPs), and a thiol-halo reaction was adopted instead of click chemistry to maintain the flexibility of phospholipids. To achieve good biocompatibility and reduce potential toxicity, structure of phosphocholine was maintained as the hydrophilic head¹⁴. In addition, as the boron-containing moiety forms liposome membranes, the internal cavity is released to carry other drugs for combination therapy.

For more information, the key criteria (tumour selectivity, treatment efficacy, etc.) of previously reported boronated liposomes have been evaluated to further highlight the advantages of boronsome (**Table R6**). Of note, the boron content of “borosome” is higher than other reported boronated liposomes and BPA (**Table R7**).

Table R6 Summary of previously reported different boronated liposomes.

No.	Compositions	Boron moiety	Max. boron in tumour (ppm)	Boron Dose (mg/kg mouse)	Tumour/blood ratio	Safety	Efficacy	Drug loading property	Imaging property
1	BoP3, DPPC, CHOL (cholesterol) and DSPE-PEG2000 (50:10:40:1) (This work)	closo -dodecarborane (at lipid tails)	93 (12 h)	39	4.2 (12 h)	No systemic toxicity or side effects have been observed	The tumour volume shrank to 1/5 after about 3 weeks (PARPi-boronsome)	✓ (Dox or PARPi)	✓ (PET-CT)
2 ⁶	DSPC, CHOL, B-lipid, DSPE-PEG-OMe (1:1:0.25:0.11). Transferrin conjugated.	nido -carborane (at lipid heads)	22 (72 h)	7.2	< 4.0 (72 h)	LD ₅₀ (48 h) = 14 mg B/kg	Average survival rate: 21 days (untreated) and 31 days (treated)	×	×
3 ⁷	DSBL, DSPC, CHOL, DSPE-PEG (0.1:0.9:1:0.11)	BSH (at lipid heads and in cavity)	35 (24 h)	15	1.5 (36 h)	At given dose no mouse died within 3 weeks	Tumour controlled within 3 weeks	×	✓ (MRI)
4 ⁸	DSPC, CHOL, and B-lipid (1:1:1)	closo -dodecarborane (at lipid heads)	N.A.	N.A.	N.A.	IC ₅₀ (24 h) = 5.6 mM (V79 Chinese hamster cells)	N.A.	×	×
5 ⁹	DSPC, CHOL, and B-lipid (1:1:1)	closo -dodecarborane (at lipid heads)	N.A.	N.A.	N.A.	IC ₅₀ (24 h) = 2 mM (V79 Chinese hamster cells)	N.A.	×	×

6 ¹⁰	DSPC, CHOL, and MAC (1:1:0.6)	TAC (in cavity) and MAC (in the middle of the bilayer)	27 (30 h)	17 /2 inj:18.6	1.4 (30 h)	No systemic toxicity or side effects have been observed	Tumour volume shrank to 12% of controlled group after 2 weeks	×	×
7 ¹¹	DSPC, CHOL, DSPE-PEG, DSPE-PEG-COOH (2:1:0.11:0.021); Transferrin conjugated	BSH (in cavity)	34 (48~72 h)	35	6.0 (72 h)	No systemic toxicity or side effects have been observed	Tumour controlled within 12 days	×	×
8 ¹²	DSPC, CHOL, B-lipid (1:1:0.6)	nido -carborane (at lipid heads) and BSH (in cavity)	32 (30 h)	11	5.3 (48 h)	No systemic toxicity or side effects have been observed	Tumour controlled within 3 weeks	×	×

Notes: (A) Structure 2, whose boron moiety were *nido*-carborane, had acute cytotoxicity which led 50 % of mice die at a dose of 14 mg B/kg weight within 48 h; (B) Structure 4 and 5 did not perform boron biodistribution or BNCT studies; (C) TAC (Na₃[*ae*-B₂₀H₁₇NH₃]) and MAC (K[*nido*-7-CH₃(CH₂)₁₅-7,8-C₂B₉H₁₁]) were in different positions of the liposomes; (D) The injection dose of boronsome was converted from the value of 500 mg (the mass of boronsome solution)/kg. N.A., not available.

Table R7 Comparison of boron percentage in different boronated liposomes and in BPA.

No.	1	2	3	4	5	6	7	8	BPA
B mass fraction/%	7.9	N.A.	N.A.	5.6	5.6	N.A.	N.A.	N.A.	5.2

Notes: Boron percentage of structure 2, 3, 6, 7 and 8 can't be precisely calculated due to limited information. N.A., not available.

2. It is not clear to the reviewer if the authors use ^{10}B -rich carborane or not. As the authors know, only 20% of the natural boron is boron-10, and hence this is a critical aspect of the experimental design. The use of ^{10}B -enriched carborane may lead to a huge cost, although it might be necessary to achieve therapeutic efficacy in the clinics. Indeed, the dose administered by the authors is extremely high (500 mg/Kg). Allometric scaling predicts a dose of ca. 40 mg/Kg in humans, which for a 80 Kg subject would be 3 grams. Is this realistic? The authors should report on the yield of all chemical reactions, to estimate how much carborane will be needed to prepare, e.g., 1 g of boronsomes.

- We thank the reviewer for this inspiring question. We used natural boron carborane due to the shortage (and import restriction) of B-10 carborane in China Mainland. We understand the reviewer's concern about the injection dose and the chemical yield to prepare boronsome. For BNCT, the interspecies allometric scaling for dose conversion from mouse to human studies is complicated, which differs from the approach that considers body surface area. For instance, ^{10}B -BPA is a powerful B-10 delivery drug that has been used in clinics, its dosage is 500 mg/kg intravenously over 3 hours¹⁵ (clinical trials: NCT04293289, NCT01173172, NCT00115453), which would be 40 grams for an 80 kg subject. Whereas 250 to 500 mg/kg of ^{10}B PBA is commonly administrated in mouse models as well¹⁶⁻²⁰.

Hence, it can be extrapolated that the clinical dosage of boronsome might be around 500 mg/kg, and no safety issue has been observed in the mice treated with boronsome at this dose according to the routine blood test (**Figure R6**) and HE (Hematoxylin and Eosin) staining (**Figure R7**). Therefore, balancing efficacy and safety, 500 mg/kg is considered reasonable in this study.

Regarding the synthesis of boronsomes, the yield of all chemical reactions is shown as in **Figure R8a**, and the overall yield is 18%. To prepare 1 g of boronsomes, 0.622 g carborane will be needed (**Figure R8b**). As the synthesis technology of carborane is fairly mature and kilogram-scale purchasing of carborane is feasible, synthesis of large amounts of boronsome is not difficult. And at the same time, we are also endeavoring to cooperate with local industries in developing ^{10}B -enriched carborane for clinical transformation in the future.

Figure R6. The routine blood test assay at day 1, day 7 and day 14 in tumour-bearing mice those were intravenously injected with 500 mg/kg boronsome.

Figure R7. Hematoxylin and Eosin (H&E) staining of hearts, livers, spleens, lungs and kidneys obtained from 4T1 tumour bearing mice. Scale bar, 200 μm .

Figure R8. Chemical synthesis and related yield of BoP-3. a. Chemical synthesis of BoPs with the yield of all chemical reactions. b. Calculation process of the amount of carborane to synthesis boronsome. Molar ratio of BoP-3, DPPC (MW: 734.04), CHOL (MW: 386.65), DSPE-PEG-2000 (MW: 2805.5) = 50:10:40:1.

3. The authors select 12 h as the time point for neutron irradiation. The long residence time in the tumor suggests that later time points could be more appropriate, to decrease Tumor-to-blood ratios. Can the authors provide dynamic time-activity-curves for all organs and the tumor-to-organ ratios? Values at 12 h are provided, but PET images should provide this info at the different time points.

- Thank the reviewer for the constructive suggestion. The values we provided at 12 h are from biodistribution studies of mice injected with ^{64}Cu Cu-NOTA-boronsome. Due to the recent COVID-19 pandemic and upcoming Winter Olympics in Beijing, the control of radionuclides has become more stringent, so it's difficult to perform time-activity curves with nuclear imaging at the moment. Nevertheless, we do agree with the reviewer that a comprehensive pharmacokinetics study is of importance to determine the timing of neutron irradiation. Therefore, we have performed a time-dependent boron concentration assay with ICP-OES, which is a more accurate method to assess the boron concentration in each organ for a long period (up to 72 h), to investigate the biodistribution and pharmacokinetics of boronsome in tumour-bearing mice.

At each time point, four tumour-bearing mice were dissected and the organs of interest were processed through microwave digestion, followed by the ICP-OES assay to determine the boron concentration. As shown in **Figure R9**, the tumour uptake of boron would continually increase till reached a plateau (up to 100 ppm) at 8-12 hours post injection. The boron concentration in blood and other major organs gradually declined after the injection, providing

high tumour-to-normal tissue ratios during 8-12 hours post injection (**Figure R10** and **R11**). Regarding the choice of 12 hours, the concentrations of boron in tumours at 8 hours and 12 hours are the peak, but the T/N ratio of each organ at 12 hours was better than that at 8 hours (**Figure R11b**), so we chose 12 hours as the timing of neutron exposure.

Figure R9. Time-dependent boron concentration in tumour, blood, lung, brain, muscle, fat, bone, heart, liver, and spleen at indicated time points after the intravenous injection of boronsome (500 mg/kg). Data are means \pm SEM ($n = 4$ for each time point).

Figure R10. Tumour-to-normal tissue ratio (T/N ratio) of boron in tumour, blood, lung, brain, muscle, fat, bone, heart, liver, and spleen at indicated time points after the intravenous injection of boronsome (500 mg/kg). Data are means \pm SEM ($n = 4$ for each time point).

Figure R11 Pharmacokinetics of boronsome. a Time-dependent boron concentration in tumour, blood, lung, brain, muscle, fat, bone, heart, liver, and spleen post once injection of boronsome (500 mg/kg, i.v.). **b** Tumour-to-normal tissue ratio (T/N ratio) of tumour, blood, lung, brain, muscle, fat, bone, heart, liver, and spleen 8 and 12 hours post once injection of boronsome (500 mg/kg, i.v.). Two-tailed unpaired Student’s t-test, **P < 0.01, ***P < 0.01. Data are means ± SEM (n = 4 mice).

4. There are a wide variety of boron-rich nanosystems reported in the literature that achieve >5% of ID/g of tumor, and hence when administered at this dose they would also achieve high concentration in the tumor. The key points here are: 1) which is the percentage of boron (¹⁰B) in the final nanosystem? and 2) how do these values compare to previous nanosystems reported in the literature? These need further consideration in the paper.

- We thank the reviewer very much for pointing out the missing experiment details. In the boronsomes, the percentage of boron is 7.9%. And the boron percentage of previous nanosystems is shown below (Table R8). We can find that (1) nanosystems are likely to possess more boron than BPA and (2) our structure has an advantage over other nanosystems. For more information, the key criteria (tumour selectivity, treatment efficacy, etc.) of previously reported boronated liposomes have been evaluated to further highlight the advantages of boronsome (Table R6).

Table R8. Comparison of boron percentage in different nanosystems and BPA.

Structure	1	4	5	BPA
B mass fraction/%	7.9	5.6	5.6	5.2

5. The authors rely in EPR effect to accumulate the boronsomes in the tumour. However, EPR effect is known to be heterogeneous and there is a large inter-subject variability. This deserves

a comment mentioning the potential limitations.

- We agree with the reviewer that the EPR effect is highly variable due to significant heterogeneity in tumour vascular permeability. The imaging-guided BNCT method shown in our study demonstrates the effectiveness of using PET imaging with radiolabelled boronsomes to measure the EPR effect. This can potentially be translated to patients, where tumours with high vascular permeability can be identified prior to making the choice of whether or not to be treated with boronsome-BNCT. We thank the reviewer to point this out. We followed the advice, and add an essential mention to explain the limitation of EPR effect and the advantages of boronsome that could enable precision medicine labelled with radioactive nuclides. Please check as below:

Noninvasive techniques that can trace boron in real-time are critical in-patient screening, treatment planning and efficacy evaluation. PET imaging has already been demonstrated powerful in imaging-guided BNCT both in preclinical and clinical studies. Meanwhile, boronsome accumulates in the tumours due to the permeability and retention (EPR) effect, which may be significantly heterogeneous in patients. Therefore, boronsomes labelled with radioactive nuclides would allow the identification before choosing whether or not to be treated with boronsome-BNCT, which could enable precision medicine.

6. The authors have selected a triple negative breast cancer model for their in vivo experiments, which is not the typical cancer type to evaluate new BNCT agents. As a proof of concept, I do not see any problem. However, TNBC is known to have a bad prognostic because it is likely to be spread at the time it is found. When the cancer is localized, 5-year survival with current therapeutic alternatives is >90%. With distant metastasis, the value drops to 12%. However, with distant metastasis BNCT would not be that useful. This deserves again a comment in the manuscript.

- We thank the reviewer for the valuable suggestion. TNBC or other breast cancers may be suitable for BNCT treatment as its location would avoid unnecessary irradiation to healthy tissues, especially other major organs. Of note, a comparison BNCT study in 4T1 tumour-bearing mice showed that the effect of BPA for the treatment of TNBC is not significant, while boronsome can make up for the deficiency (**Figure R12**). This result may highlight the potential of boronsome-BNCT to treat TNBC. In addition, with regard to distant metastasis, several recent works suggest that BNCT may have distant effects^{21,22}, and the potential of boronsome-BNCT on metastatic TNBC may be further studied in the future.

Figure R12. Tumour volumes (n = 4) of each group of mice treated with Boronsome+N and BPA+N. Paired Student's t-test, *P < 0.05, Data are means ± SEM.

7. Characterisation of labelled liposomes and stability: Fig. S7 is very poor. Are these TLC profiles? If yes, the X axis is not correct (should be distance). Also, at long times it seems that a second peaks tends to appear on the right... but the curves are cut... the authors need to clarify this.

- We apologize for the unclear presentation in the previous submission. About X-axis in these radio-TLC assay, the paper tape moves at a constant speed, so time could represent distance, Distance (mm) = Time (min) x 1 mm/s. We put a lot of effort into getting the minimal amount of [⁶⁴Cu]Cu just enough to do this assay and then refurbished the radio-TLC assay. The results showed that radiolabelled boronsome is stable in 50% FBS for at least 24 hours (**Figure R13**).

Figure R13. In vitro stability assay of [⁶⁴Cu]Cu-NOTA-boronsome. Radio TLC chromatography of free Cu-64 and [⁶⁴Cu]Cu-NOTA-boronsome after incubation in serum at 37 °C for 24 h (eluant: 0.05 mM EDTA).

8. For the combination therapies, which is the dose of the drug administered to the animals? It is always 500 mg/Kg in boronsomes + the amount of drug? It is important to clarify which is the amount of boron and drug administered in each case (all groups).

- We thank the reviewer for the constructive suggestion, and apologize for the unclear presentation in the previous submission. In this work, the injections of combination therapies (including boronsome+Dox and boronsome+Olaparib) were all conducted at the dose of 500 mg/kg in boronsome and drug (the molar ratio of drug to boronsome was 1:8). The feasibility of this specific molar ratio has been tested in other liposomal drug delivery systems. For instance, light-triggered DOPC-containing porphyrin–phospholipids liposomes have been designed for Dox delivery treating human pancreatic xenograft²³. And the liposome with a molar ratio of 1:8 (drug to lipid) showed excellent loading and delivery efficacy. Therefore, we adopted this molar ratio to prepare our boronsomes.

9. Citations are quite old most of them. It is true that a huge portion of the work related to BNCT and liposomes was performed a couple of decades ago. However, there are recent works related to boron-rich nanomaterials that have been ignored. Both for citation and for discussion

with comparative purposes.

- We thank the reviewer for this valuable advice. As suggested, we have added a discussion of the advantages of borosome when compared to previously reported nanosystems with similar compositions in the revised manuscript. The related references have been added accordingly.

10. The TOC file is not representative of the scope of the work and needs improvement

- As suggested, a new schematic figure (scheme R1d) has been added for better presentation in the revised manuscript.

Scheme R1. Schematic illustration of boronated liposomes for BNCT

Reviewer #4

This work presents a clever strategy to prepare liposomes that are able to promote simultaneous delivery of boron and chemotherapy drugs to brain tumors. The innovative strategy stands out for the use of a boron-modified lipid molecule that could integrate the lipid bilayer of a liposome structure and therefore, instead of being leaked from the liposome as usually is found with this type of nanoparticle, the boron is released from this bilayer. The authors explored several applications and proved that the borosome is useful as drug-delivery platform as well as theranostic platform since it could be also loaded with radioactive chemicals.

In my opinion this work should be considered to be published in Nature Communications but some suggestions should be considered.

- We thank the reviewer for the positive comments on both the innovative chemical design and the applications in cancer treatment parts of our work.

1. Although the application of borosome as theranostic platform has been shown, in vitro cytotoxicity as well as in vivo toxicity data are missing with the radioactive-labeled borosomes.

- We thank the reviewer for this constructive advice. As suggested, we have performed a systematic evaluation to evaluate the safety issues of radioactive-labeled boronsomes, both *in vitro* and *in vivo*.

Figure R14. Cell viability assay of 4T1 cells under various dosages of [⁶⁴Cu]Cu-NOTA-boronsome after 24 h assessed with a CCK-8 assay (n = 3).

Figure R15. Evaluation of potential side effect of $[^{64}\text{Cu}]\text{Cu-NOTA-boronsome}$ in mice. a. The routine blood test assay in tumour-bearing mice those were intravenously injected with $[^{64}\text{Cu}]\text{Cu-NOTA-boronsome}$ at day 7. **b.** Hematoxylin and Eosin (H&E) staining assay of hearts, livers, spleens, lungs and kidneys obtained from 4T1 tumour-bearing mice treated by PBS and $[^{64}\text{Cu}]\text{Cu-NOTA-boronsome}$, respectively. Scale bar, 200 μm .

1) For *in vitro* cytotoxicity study, we picked a dosage (0.11 μCi , 2×10^4 cells) that far exceeds 100-fold the activity of tumour region (5.5 μCi , about 10^8 cells), and exceed (22-fold) the activity of liver region (75 μCi , about 3×10^8 cells), which has a highest uptake of $[^{64}\text{Cu}]\text{Cu-NOTA-boronsome}$ according to the PET study. As shown in **Figure R14**, cell viability was not

affected even at the highest dose, which showed the excellent *in vitro* safety profile for radioactive-labeled boronsomes.

2) For *in vivo* toxicity study, we have performed the routine blood test and HE staining in tumour-bearing mice treated by PBS and [⁶⁴Cu]Cu-NOTA-boronsome, respectively (**Figure R15**). Mice were treated by intravenous injection with 7.4 MBq [⁶⁴Cu]Cu-NOTA-boronsome, then were sacrificed on day 7. The routine blood test showed no significant cytotoxicity. And the blood biochemical test indicated that all liver and kidney function parameters were within the normal range (**Figure R15a**). As shown in **Figure R15b**, no abnormalities have been observed in major organs including the heart, liver, spleen, lung and kidney.

2. In page 4 it was mentioned the absence of toxicity of radiolabeled boronsomes but even if the boronsomes were selectively accumulated in the tumor tissue, it was possible to find them in normal tissue but there was no histological studies in these tissues, as it was performed with DOX-loaded boronsomes.

- As suggested, we have performed the routine blood test and HE staining at 7 days post injection of ⁶⁴Cu-NOTA-boronsome as we replied in comments #1 (**Figure R15**), the histological of normal tissues (including heart, liver, spleen, lung and kidney) showed no obvious abnormalities.

In addition, the potential side-effect of 500 mg/kg boronsome has also been evaluated in tumour-bearing mice. As shown below, no safety issue has been observed in the mice treated with boronsome at this dose according to the routine blood test (**Figure R16**) and HE (Hematoxylin and Eosin) staining (**Figure R17**).

Figure R16. The routine blood test assay at day 1, day 7 and day 14 in tumour-bearing mice those were intravenously injected with 500 mg/kg boronsome.

Figure R17. Hematoxylin and Eosin (H&E) staining of hearts, livers, spleens, lungs and kidneys obtained from 4T1 tumour bearing mice. Scale bar, 200 μm .

3. It was evident the lack of recently published research works among the consulted references. The data should be commented comparing the recent results obtained by similar research works. There was only 2 references from 2020 and only 1 from 2021.

- We thank the reviewer for this valuable advice. As suggested, we have added a discussion of the advantages of boronsome when compared to previously reported nanosystems with similar compositions in the revised manuscript. The related references have been added accordingly.

Some minor comments:

4. In the abstract "unsatisfied boron delivery agents". Please complete the sentence by pointing delivery to "brain tumor" or other suggestion the authors deem appropriated.

- Thank the reviewer very much for the valuable suggestion. We followed the above advice and rephrase our related description as follows:

"but its clinical applications have been hindered by boron delivery agents with low in vivo stability, poor biocompatibility, or limited application of combinational modalities."

5. Abstract 4th line: "boronsome, a carboranyl-phosphatidylcholine based lipid bilayer for the first time". It would be properly called liposome instead of lipid bilayer because it is not a "sandwich bilayer", it is a spherical nanoparticle as shown in the TEM images.

- We agree with the reviewer and have replaced "lipid bilayer" with "liposome" in the revised manuscripts. We have made a thorough check and highlighted the changes in the revision copy.

6. Abstract 6th line: "treated by boronsome". It should be correct to "treated with".

- Thanks, we have made the correction accordingly in the revised manuscript.

7. First page, last line before scheme 1: "Lack of non-invasive imaging techniques to explore in vivo biodistribution of boron agents. It could be better explained that the problem is not only the missing technique but absence of suitable properties of boron agents to allow their detection by imaging techniques.

- Thanks to the reviewer for the constructive suggestion. We followed the above advice and rephrase our related description as follows:

"Absence of suitable properties to allow their detection by non-invasive imaging techniques to explore in vivo biodistribution of boron agents to ensure the accuracy of neutron irradiation and to improve efficacy."

8. First line, page 2: correct "to developed" to "develop".

- Thanks, we have corrected the related description in the revised manuscript.

9. First line, page 2, second paragraph: how the stability of boronsome was measured? It was implicit that was by dynamic light scattering (DLS) measurement. However, it was not mentioned the technique as well as it was not mentioned the measurement of Hydrodynamic diameter. It is crucial to characterize this property if the mentioned "size" and polydispersity index was not measured by DLS.

- We apologized for missing this information and the experimental details have been added to the "Methods" part in the revised manuscript, as follows:

Intensity-average hydrodynamic diameters and size distributions (PDI) were measured by Dynamic laser light scattering (DLS) which were conducted on a Zetasizer Nano ZS system (Malvern Instruments Ltd, England). The scattered light was collected at a fixed angle of 173° for a duration of ~5 min. All data were averaged over three consecutive measurements.

10. Values of T/N ratio was not appropriately reported since the deviation/error has excessive significant digits. For example, the value 36.9 ± 1.41 should be corrected to 37 ± 1 .

- We thank the reviewer for his/her suggestion. In the revised manuscript, we have adjusted the format of deviation/error in the revised manuscript.

11. Page 3, 3rd paragraph: "naive" should be replaced by "unloaded".

- We agree with the reviewer and have replaced "naive" with "unloaded" in the revised manuscripts. We have made a thorough check and highlighted the changes in the revision copy.

12. Page 3, 3rd paragraph: It has to be further explained why "electric neutralized" can be considered an evidence of dox loading. In fact, it has to be correctly called "lower Zeta potential". It has to be kept in mind that Zeta potential is an indicative of surface charge and therefore if after DOX loading the Zeta potential was decreased in modulus, this is an evidence that DOX is close to the surface instead of encapsulated.

- We thank the reviewer for the constructive suggestion. Dox has been encapsulated in the boronsome using an ammonium sulfate gradient loading procedure which is very classic. We have corrected the related description in the revised manuscript, and added an explanation as follows:

"Zeta-potential of liposomes, boronsomes and Dox-boronsomes were measured (Fig. 2e), which expresses the potential difference between the dispersion medium and the stationary layer of fluid attached to double-layer properties to characterize a realistic magnitude of

surface charge²⁴.”

Reference

- 1 Heber, E. M. *et al.* Therapeutic efficacy of boron neutron capture therapy mediated by boron-rich liposomes for oral cancer in the hamster cheek pouch model. *Proc. Natl. Acad. Sci. U. S. A.* **111**, 16077-16081 (2014).
- 2 Nakamura, H., Miyajima, Y., Takei, T., Kasaoka, S. & Maruyama, K. Synthesis and vesicle formation of a nido-carborane cluster lipid for boron neutron capture therapy. *Chem. Commun. (Cambridge, U. K.)* **17**, 1910-1911 (2004).
- 3 Bregadze, V. I. *et al.* Boron-Containing Lipids and Liposomes: New Conjugates of Cholesterol with Polyhedral Boron Hydrides. *Chemistry* **26**, 13832-13841 (2020).
- 4 Li, T., Hamdi, J. & Hawthorne, M. F. Unilamellar liposomes with enhanced boron content. *Bioconjugate Chem.* **17**, 15-20 (2006).
- 5 Imahori, Y. *et al.* Fluorine-18-labeled fluoroboronophenylalanine PET in patients with glioma. *J. Nucl. Med.* **39**, 325-333 (1998).
- 6 Miyajima, Y. *et al.* Transferrin-Loaded nido-Carborane Liposomes: Tumor-Targeting Boron Delivery System for Neutron Capture Therapy. *Bioconjugate Chem.* **17**, 1314-1320 (2006).
- 7 Koganei, H. *et al.* Development of High Boron Content Liposomes and Their Promising Antitumor Effect for Neutron Capture Therapy of Cancers. *Bioconjugate Chem.* **24**, 124-132 (2013).
- 8 Justus, E. *et al.* Synthesis, liposomal preparation, and in vitro toxicity of two novel dodecaborate cluster lipids for boron neutron capture therapy. *Bioconjugate Chem.* **18**, 1287-1293 (2007).
- 9 Schaffran, T. *et al.* Dodecaborate cluster lipids with variable headgroups for boron neutron capture therapy: Synthesis, physical-chemical properties and toxicity. *J. Organomet. Chem.* **694**, 1708-1712 (2009).
- 10 Kueffer, P. J. *et al.* Boron neutron capture therapy demonstrated in mice bearing EMT6 tumors following selective delivery of boron by rationally designed liposomes. *Proc. Natl. Acad. Sci. U. S. A.* **110**, 6512 (2013).
- 11 Maruyama, K. *et al.* Intracellular targeting of sodium mercaptoundecahydrododecaborate (BSH) to solid tumors by transferrin-PEG liposomes, for boron neutron-capture therapy (BNCT). *J. Control Release* **98**, 195-207 (2004).
- 12 Feakes, D. A., Shelly, K. & Hawthorne, M. F. Selective boron delivery to murine tumors by lipophilic species incorporated in the membranes of unilamellar liposomes. *Proc. Natl. Acad. Sci. U. S. A.* **92**, 1367-1370 (1995).
- 13 Luderer, M. J. *et al.* Thermal Sensitive Liposomes Improve Delivery of Boronated Agents for Boron Neutron Capture Therapy. *Pharm. Res.* **36**, 144 (2019).
- 14 Zhang, Y., Sun, C., Wang, C., Jankovic, K. E. & Dong, Y. Lipids and Lipid Derivatives for RNA Delivery. *Chem. Rev.* **121**, 12181-12277 (2021).
- 15 Suzuki, M. *et al.* Boron neutron capture therapy outcomes for advanced or recurrent head and neck cancer. *J. Radiat. Res.* **55**, 146-153 (2014).
- 16 Kankaanranta, L. *et al.* Boron neutron capture therapy (BNCT) followed by intensity modulated chemoradiotherapy as primary treatment of large head and neck cancer with intracranial involvement. *Radiother. Oncol.* **99**, 98-99 (2011).
- 17 Yamamoto, T. *et al.* Boron neutron capture therapy for newly diagnosed glioblastoma. *Radiother. Oncol.* **91**, 80-84 (2009).

- 18 Kankaanranta, L. *et al.* Boron neutron capture therapy in the treatment of locally recurred head-and-neck cancer: final analysis of a phase I/II trial. *Int. J. Radiat. Oncol., Biol., Phys.* **82**, e67-75 (2012).
- 19 Wang, L. W. *et al.* BNCT for locally recurrent head and neck cancer: Preliminary clinical experience from a phase I/II trial at Tsing Hua Open-Pool Reactor. *Appl. Radiat. Isot.* **69**, 1803-1806 (2011).
- 20 Chanana, A. D. *et al.* Boron neutron capture therapy for glioblastoma multiforme: interim results from the phase I/II dose-escalation studies. *Neurosurgery* **44**, 1182-1192; discussion 1192-1183 (1999).
- 21 Trivillin, V. A. *et al.* Abscopal effect of boron neutron capture therapy (BNCT): proof of principle in an experimental model of colon cancer. *Radiat. Environ. Biophys.* **56**, 365-375 (2017).
- 22 Trivillin, V. A. *et al.* Evaluation of local, regional and abscopal effects of Boron Neutron Capture Therapy (BNCT) combined with immunotherapy in an ectopic colon cancer model. *Br. J. Radiol.* **94**, 20210593 (2021).
- 23 Luo, D. *et al.* Rapid Light-Triggered Drug Release in Liposomes Containing Small Amounts of Unsaturated and Porphyrin-Phospholipids. *Small (Weinheim an der Bergstrasse, Germany)* **12**, 3039-3047 (2016).
- 24 Honary, S. & Zahir, F. Effect of zeta potential on the properties of nano-drug delivery systems-a review (Part 1). *Trop. J. Pharm. Res.* **12**, 255-264 (2013).

REVIEWERS' COMMENTS

Reviewer #1 (Remarks to the Author):

I read the revised manuscript and found that the authors have addressed most of my comments. However, the following issues still remain questionable:

- 1) Regarding in vivo BNCT experiments, Dox-boronsome controlled the tumor growth more effectively compared to boronsome under the neutron irradiation for 20 days (Fig. 4C). However, the survivals of mice treated with both agents were the same at 100% for 21 days. How was it for longer observations?
- 2) The revised Supporting Information lacks spectral data; full identification data for new compounds should be provided, including not only ^1H NMR, but also ^{13}C NMR, high-resolution mass spectra and mp.

Reviewer #2 (Remarks to the Author):

The authors have done a great job in clarifying all my doubts and concerns. The quality of the work has significantly improved. I think the manuscript is suitable for publication in its current form. Just one minor to be corrected:

- Radioactivity should be expressed in Bq, not in "curies".

Reviewer #4 (Remarks to the Author):

The authors made an expressive upgrade in the manuscript. Crucial in vivo and in vitro assays were added in order to prove the low toxicity of the boronsome to non tumoral tissues. Also, the authors added important discussion regarding more recent works than the ones that have been commented in the previous version. Missing information regarding the characterization was also included. Another improvement that is noteworthy is the additional comparative analysis between boronsome and BPA. This analysis could improve the quality of the work since it evidenced the advantages of the claimed material.

RESPONSES TO THE COMMENTS

Please kindly find our point-by-point response below.

Reviewer #1 (Remarks to the Author):

I read the revised manuscript and found that the authors have addressed most of my comments. However, the following issues still remain questionable:

1) Regarding in vivo BNCT experiments, Dox-boronsome controlled the tumor growth more effectively compared to boronsome under the neutron irradiation for 20 days (Fig. 4C). However, the survivals of mice treated with both agents were the same at 100% for 21 days. How was it for longer observations?

- We thank the reviewer for this valuable advice. Though remarkable suppressions on tumour growth were observed in both BNCT and BNCT with Dox-boronsome groups, the following observations showed that the BNCT with Dox-boronsome can be more effective: 1) The tumours treated by BNCT with Dox-boronsome shrank while BNCT only group increased. The average tumour volumes of BNCT and BNCT with Dox-boronsome groups on Day 21 is 1.6-fold and 0.5-fold compared to those on Day 0, respectively; 2) Of note, 4 of 9 tumours treated by BNCT with Dox-boronsome were eradicated within three weeks, in comparison, only 2 of 9 tumours were cured with BNCT only. Therefore, we would suggest that the experimental purpose has been achieved that BNCT with Dox-boronsome suppressed the tumor growth more effectively, and it may not necessary to observe the survival rate of mice for a longer period.

2) The revised Supporting Information lacks spectral data; full identification data for new compounds should be provided, including not only ^1H NMR, but also ^{13}C NMR, high-resolution mass spectra and mp.

- We thank the reviewer for this suggestion. The high-resolution mass spectra of compounds 20–23 have been performed and described in the original SI. Though difficult, we have also been trying very hard to assess the high-resolution mass spectra of compounds 16–19. The high-resolution mass spectra of compounds 16–19 are shown below. The mass spectra of these compounds feature multiple isotope peaks (each compound contains 10 boron atoms, the ratio of $^{10}\text{B}/^{11}\text{B}$ is 1:4).

Peking University Mass Spectrometry Sample Analysis Report

Analysis Info

Analysis Name FTMS-22030070_Pos_20220307_000005.d Acquisition Date 3/7/2022 10:32:16 AM
 Sample 4C-COOH Instrument Bruker Solarix XR FTMS
 Comment Operator Peking University

Figure R1 High-resolution mass spectrum of compound 16.

Peking University Mass Spectrometry Sample Analysis Report

Analysis Info

Analysis Name FTMS-22030071_Pos_20220307_000006.d Acquisition Date 3/7/2022 10:23:38 AM
 Sample 8C-COOH Instrument Bruker Solarix XR FTMS
 Comment Operator Peking University

Figure R2 High-resolution mass spectrum of compound 17.

Peking University Mass Spectrometry Sample Analysis Report

Analysis Info

Analysis Name FTMS-22030072_Pos_20220307_000001.d
Sample 12CCOOH
Comment

Acquisition Date 3/7/2022 10:12:34 AM
Instrument Bruker Solarix XR FTMS
Operator Peking University

Figure R3 High-resolution mass spectrum of compound 18.

Peking University Mass Spectrometry Sample Analysis Report

Analysis Info

Analysis Name FTMS-22030073_Pos_20220307_000005.d
Sample 16C-COOH
Comment

Acquisition Date 3/7/2022 10:07:10 AM
Instrument Bruker Solarix XR FTMS
Operator Peking University

Figure R4 High-resolution mass spectrum of compound 19.

Figure R1–4 have been added to the revised SI as Figure S24–27.

Reviewer #2 (Remarks to the Author):

The authors have done a great job in clarifying all my doubts and concerns. The quality of the work has significantly improved. I think the manuscript is suitable for publication in its current form. Just one minor to be corrected:

- Radioactivity should be expressed in Bq, not in "curies".

- Thank the reviewer for the constructive suggestion. The unit "Ci" have been revised to "Bq" according to unit conversion rules in the final submission.

Reviewer #4 (Remarks to the Author):

The authors made an expressive upgrade in the manuscript. Crucial in vivo and in vitro assays were added in order to prove the low toxicity of the boronsome to non tumoral tissues. Also, the authors added important discussion regarding more recent works than the ones that have been commented in the previous version. Missing information regarding the characterization was also included. Another improvement that is noteworthy is the additional comparative analysis between and BPA. This analysis could improve the quality of the work since it evidenced the advantages of the claimed material.

- We thank the reviewer for the enthusiastic comments.